# MITIGATING THE CURSE OF DETAIL: SCALING ARGUMENTS FOR FEATURE LEARNING AND SAMPLE COMPLEXITY

Noa Rubin[1], Orit Davidovich[2], and Zohar Ringel[1]

[1]Racah Institute of Physics, Hebrew University, Jerusalem, Israel
[2]IBM Research, Haifa, Israel.

## ABSTRACT

Two pressing topics in the theory of deep learning are the interpretation of feature learning (FL) mechanisms and the determination of implicit bias of networks in the rich regime. Current theories of rich FL often appear in the form of high-dimensional non-linear equations, which require computationally intensive numerical solutions. Given the many details that go into defining a deep learning problem, this analytical complexity is a significant and often unavoidable challenge. Here, we propose a powerful heuristic route for predicting the data and width scales at which various patterns of FL emerge. This form of scale analysis is considerably simpler than such exact theories and reproduces the scaling exponents of various known results. In addition, we make novel predictions on complex toy architectures, such as three-layer non-linear networks and attention heads, thus extending the scope of first-principle theories of deep learning.

## 1    INTRODUCTION

There is a clear need for a better theoretical understanding of deep learning. However, efforts to construct such theories inevitably suffer from a "curse of details". Indeed, since any choice of architecture, activation, data measure, and training protocol affects performance, finding a theory with true predictive power that accurately accounts for all those details is unlikely. One workaround is to focus on analytically tractable toy models, an approach that can often uncover interesting fundamental aspects. However, analytical tractability is a fragile, fine-tuned property; thus, a large explainability gap remains between such toy models and more complex data/architecture settings.

An alternative approach focuses on scaling properties of neural networks, which appear more robust. Two well-established examples are empirically predicting network performance by extrapolating learning curves using power laws Kaplan et al. (2020); Hestness et al. (2017), and providing theory-inspired suggestions for hyperparameter transfer techniques Yang et al. (2022); Bordelon et al. (2023). Indeed, it is often the case Cardy (1996) that predicting scaling exponents is easier than predicting exact or approximate behaviors. As a simple toy model of this, consider the integral $\int_{-\infty}^{\infty} dx g(x/P)$. While $g(\cdot)$ needs to be fine-tuned for exact computations, a change of variable reveals a robust linear scaling with $P$ for any $g(\cdot)$.

This work focuses on scaling properties of feature learning (FL). This phenomenon, and, more generally, interpretability, has been studied extensively both from the practical and theoretical sides. On the practical side, mechanistic interpretability Bereska and Gavves (2024) has provided us with statistical explanations for why some predictions are made and the underlying decision mechanisms. On the theory side, kernel approaches Bordelon and Pehlevan (2022); Aitchison (2021) Saad and Solla type approaches Saad and Solla (1995); Arnaboldi et al. (2023); Bietti et al. (2022) (and their Bayesian counterparts Cui (2025)) allow us to solve simple non-linear teacher-student networks in the rich regime.

This work examines feature learning through a Bayesian lens, where it is often defined as a deviation from the Gaussian Process (GP) prior. This deviation has been shown to manifest in several distinct

Figure 1: Logical flow of sample complexity derivation. **Bounds: (i)** we lower bound the test MSE by $(1 - A_f)^2$, where $A_f$ is the alignment of output and target; **(ii)** we establish an upper bound on the probability to observe good learning (i.e., strong alignment $A_f \geq \alpha \approx 1$) in the posterior using the negative-log-probability of the rare event of good learning in the prior; **(iii)** we define the energy $E(\alpha)$ as the negative log-Chernoff bound on the *prior* probability of successful learning (rate function); **(iv)** we leverage $E(\alpha)$ to bound the minimal sample size necessary for good learning in the *posterior*; **Approximations: (v)** Since the exact bound $E(\alpha)$ is intractable, we derive a variational approximation using kernel-adaptation techniques to provide an explicit formula (Sec. 4). **Heuristics: (vi)–(vii)** we propose "feature learning patterns" as heuristic variational candidates, selecting the pattern $q_*$ that minimizes the approximation and utilizing heuristic scaling relations to model how feature amplification propagates through downstream layers (Sec. 5).

regime dependent ways, including: (i) updates to the kernel while the distribution remains Gaussian Li and Sompolinsky (2021); Rubin et al. (2025); Ariosto et al. (2022); Seroussi et al. (2023b); Aitchison (2019); (ii) the distribution changes from zero mean-Gaussian to a mixture of Gaussians Rubin et al. (2024); and (iii) the specialization of specific neurons Barbier et al. (2025); as well as combinations of the above Ringel et al. (2025); Meegen and Sompolinsky (2024). However, the frameworks capturing these FL effects are often mathematically cumbersome and restricted to shallow networks. Even when extended to deep architectures Barbier et al. (2025); Li and Sompolinsky (2020), their computational complexity prevents the derivation of intuitive scaling laws. Ultimately, this difficulty in computation compounds with the context-specificity of these mechanisms, complicating unified comparison of scaling laws across varied architectural settings.

In this work, we introduce a novel heuristic framework addressing the challenging task of making first-principles predictions on sample complexity and feature learning effects in Bayesian NNs. We consider Bayesian NNs as these are a standard proxy for SGD-trained networks at equilibrium Mingard et al. (2020); Mandt et al. (2017). The heuristic nature of our approach (see Fig. 1 for a schematic overview), allows us to bypass otherwise highly complex analysis and extract insights using simple pen-and-paper calculations, resulting in non-trivial predictions across a broad range of architectures and settings. We indeed validate our framework for two- and three-layer fully connected networks (using ReLU and Erf activations), attention heads, and convolutional neural networks, covering both teacher-student, regression, and classification tasks. Across these diverse settings, our main results are:

1. **Predicting feature learning patterns:** We predict which of the FL phenomena from the literature discussed above will emerge in different architectures and across varying scales of architectural variables, such as input dimension, layer width, regularization, and parametrization choices such as mean-field versus standard scaling. We map transitions between these patterns and derive the explicit scaling laws governing their strength.
2. **Predicting sample complexity scale:** We determine the scaling behavior of $P_*$, the threshold sample size at which learning is possible, as a function of the architectural variables and parametrization choices.

## 2 SETUP

We consider here several types of feedforward networks, but, for the sake of clarity, we illustrate the main derivation on deep fully connected networks (FCNs) and, later, when we analyze specific problems, augment it for convolutional neural networks (CNNs) as needed. Our FCNs are defined

by

$$f(x) = \sum_{i=1}^{N_{L-1}} w_i^L \sigma(h_i^{L-1}(x)), \quad \text{where } h_i^{l>1}(x) = \sum_{j=1}^{N_{l-1}} W_{ij}^l \sigma(h_j^{l-1}(x)), \ h_j^1(x) = \sum_{k=1}^{d} W_{jk}^1 x_k, \tag{1}$$

where $\sigma$ can be any activation function, and we refer to $h_i^l$'s as *pre-acitvations*. We consider Bayesian neural networks, as Bayesian descriptions are a commonly used proxy for network behavior after long-time stochastic training Wilson and Izmailov (2020); Wilson (2020); Naveh et al. (2021). Alternatively, they represent an exact solution to Langevin dynamics with weight decay Welling and Teh (2011b). We denote the target function by $y$ and the training sample size by $P$, and assume training with Mean Squared Error (MSE) loss. The quadratic weight decay for each layer $l$ is set to $\kappa N_{l-1}/\sigma_l^2$, where $\kappa$ is the ridge parameter and $N_0 = d$ is the input dimension. This choice of weight decay results in a Gaussian prior distribution for the weights given by $W_{i,j}^l \sim \mathcal{N}(0, \sigma_l^2/N_{l-1})$ for $i = 1, .., N_l$, $j = 1, ..., N_{l-1}$. The possible outputs $f$ of such a network, given $y$ and $P$, are then distributed according to the posterior:

$$\pi(f \mid y, P) = \frac{1}{Z} \exp\left(-\frac{1}{2\kappa} \sum_{\nu=1}^{P} [f(x_\nu) - y(x_\nu)]^2\right) p_0(f), \tag{2}$$

where $Z$ is the normalization constant, $\{x_\nu\}_{\nu=1}^P$ is the training dataset of size $P$, and $p_0(f)$ is the prior defined as $p_0(f) = \int d\Theta p(\Theta) \delta[f - f_\Theta]$, determined by the weight decay. Here, $\Theta$ is the collection of all network weights, and $p(\Theta)$ corresponds to the density of the prior weight distribution, which we take to be Gaussian with a diagonal covariance, representing quadratic weight decay, and $f_\Theta$ is the network architecture with weights $\Theta$. We further set $p(\Theta)$ such that pre-activations are all $\mathcal{O}(1)$ under the prior He et al. (2015). For classification tasks, see App. A.2. As a measure for learning, we consider an observable, which we refer to as *alignment*, given by

$$A_f := \langle f, y \rangle / \langle y, y \rangle, \tag{3}$$

where $\langle g, h \rangle = \int d\mu_x g(x) h(x)$ is the functional inner product, and $d\mu_x$ is some test measure, which, conveniently, does not need to be the measure from which the training set was drawn. We similarly define $\langle g, K, h \rangle = \int d\mu_x d\mu_{x'} g(x) K(x, x') h(x')$ for any kernel $K$. Alignment represents the extent to which the network learns a function that is proportional to the target. It bounds the test MSE via the Cauchy–Schwarz inequality $\int d\mu_x (f(x) - y(x))^2 \geq \langle y, y \rangle (A_f - 1)^2$. Having $A_f \approx 1$ is thus a necessary condition for successful learning.

## 3 ALIGNMENT AND SAMPLE COMPLEXITY

We turn to analyze sample complexity via an upper bound on the probability of finding $A_f \geq \alpha$ for $\alpha \approx 1$. We begin with a theoretical bound on the posterior that mainly depends on the chance that a random network, chosen from the prior, produces an alignment of at least $\alpha$. We denote the prior and posterior alignment probabilities by $\Pr_{p_0}[A_f \geq \alpha]$ and $\Pr_\pi[A_f \geq \alpha]$ respectively. Following simple arguments (see App. A and App. A.2 for generalization to classification.), we obtain the following bound on the log posterior [1]

$$\log(\Pr_\pi[A_f \geq \alpha]) < Pk/(2\kappa) + \log(\Pr_{p_0}[A_f \geq \alpha]), \tag{4}$$

where $k = P^{-1} \sum_{\nu=1}^P \mathbb{E}_{p_0}[(f(x_\nu) - y(x_\nu))^2]$ is the only training-set dependent quantity and is generally of order one. The Bayesian interpretation of successful learning is having $\Pr_\pi[A_f \geq \alpha] \approx \mathcal{O}(1)$. [2]. Since a random network is unlikely to achieve strong alignment, $\log \Pr_{p_0}[A_f \geq \alpha]$ would typically be highly negative for large $\alpha$. Therefore, a sufficiently large data term is required to cancel this effect. Explicitly,

$$P \gtrsim -2\kappa \log \Pr_{p_0}[A_f \geq \alpha]/k. \tag{5}$$

Thus, up to the ridge parameter and the $\mathcal{O}(1)$ factor $k$, depending on the training set, the log probability of prior alignment with the target lower bounds the sample complexity. Here, it is worth

---

[1] See also Lavie and Ringel (2025), for a similar data-agnostic bound in the context of lazy learning.

[2] More concretely, we can assume some tolerance $\epsilon$ and then $\pi(A_f \approx 1) = (\epsilon)^{-1} \int_0^\epsilon d\epsilon \pi(A_f = 1 - \epsilon)$

noting that the bound becomes tight when overfitting effects are small, which is typically the case for $k/\kappa \sim \mathcal{O}(1)$. Taking $\kappa \to 0$ encourages overfitting (though often benignly Bartlett et al. (2020)) and trivializes this bound. We conjecture that, in this case, $\kappa$ should be kept $\mathcal{O}(1)$ based on the effective ridge treatment Canatar et al. (2021b); Cohen et al. (2021); Bartlett et al. (2020). Establishing this conjecture is outside the scope of this work. From a PAC-Bayesian perspective, an analogous bound would require $P$ to be much larger than the KL-divergence between the prior and posterior[3] (e.g., McAllester (1999)). More recently, prior-posterior relations have been studied in the context of complex Boolean functions Mingard et al. (2025).

Following the Chernoff inequality, we can find an upper bound for the probability (and a lower bound for $P$) via

$$P \geq -2\kappa/k \, \log \mathrm{Pr}_{p_0}\left[A_f \geq \alpha\right] \geq 2\kappa/k \, E(\alpha), \quad E(\alpha) = -\log \inf_{t>0} e^{-t\alpha} \mathbb{E}_{p_0}\left[e^{tA_f}\right]. \quad (6)$$

Where we refer to $E(\alpha)$ as the *energy*. We can thus express the minimal sample size necessary for learning, $P_*$, through the energy as $P_* \propto E(\alpha)$. In App. B, we provide an asymptotically exact solution for $E(\alpha)$, and compute it explicitly for a two layer network. We also argue and demonstrate that our bound is inherently tied to FL. Indeed, a network sampled from the prior that achieves such alignment is a statistical outlier, driven by the emergence of an internal structure which mimics FL (see also Fig. 2). Nevertheless, such a direct LDT approach is computationally prohibitive in most cases of interest. We therefore introduce a heuristic LDT-based method for evaluating $P_*$. This method not only enables predicting the scaling of $P_*$ but also the FL effects that lead to successful learning.

In this section, we adopt a variational approach to estimating $P_*$ by comparing different modes of feature learning [4] under a certain loss (see (11) below). While many approaches predict different FL mechanisms Pacelli et al. (2023); Fischer et al. (2024); Meegen and Sompolinsky (2024); Buzaglo et al. (2025); Li and Sompolinsky (2021); Seroussi et al. (2023c); Rubin et al. (2025; 2024); Barbier et al. (2025), they are often case-dependent, highly detailed, and complex. Thus, we propose a method that abstracts key FL mechanisms from these frameworks into distinct, comparable patterns.

## 4 VARIATIONAL ANALYSIS

Our next objective is to make the sample complexity bound tractable. This requires estimating the prior probability term, $\mathrm{Pr}_{p_0}\left[A_f \geq \alpha\right]$, for alignments $\alpha \approx 1$. As a first step, we simplify this by relating the cumulative distribution function to the probability density denoted by $p_{A_f}(\alpha)$. As shown in App. A.3, for large alignments, we have $E(\alpha) \approx -\log p_{A_f}(\alpha)$. This allows us to re-express $P_*$ in terms of the density: $P_* = -2\kappa \log p_{A_f}(\alpha)/k$. However, computing $p_{A_f}(\alpha)$ directly remains intractable. We therefore turn to a variational approach to estimate it. As explained in the next section, we wish to express the variational probability density in terms of pre-activations $h$ (1). Accordingly, in App. C.1, we follow standard statistical mechanics techniques to express this density as

$$p_{A_f}(\alpha) = \int \mathcal{D}h \, \mathcal{N}\left(\alpha \mid 0, \langle y, \tilde{K}_{L-1}, y\rangle\right) \prod_{l=1}^{L-1} \prod_{i=1}^{N_l} \mathcal{N}\left(h_i^l \mid 0, \tilde{K}_{l-1}\right) \quad (7)$$

Where for each $l$, the kernels $\tilde{K}_{l-1}$ themselves depend on the preactivations of layer $l-1$, and similarly the fluctuations in $\alpha$ depend on the preactivations of the penultimate layer, given by

$$\tilde{K}_{l>0}(x, x') = \frac{\sigma_{l+1}^2}{N_l} \sum_{i=1}^{N_l} \sigma\left(h_i^l(x)\right) \sigma\left(h_i^l(x')\right), \quad K_0(x, x') = \frac{\sigma_1^2}{d} x \cdot x' \quad (8)$$

---

[3]Following the data-processing-inequality, one can lower-bound the KL-divergence between the full prior and posterior probabilities by the KL-divergence of a coarser probability of an $A_f \geq \alpha$ event in the prior and posterior. The latter KL divergence is given by $-\log \mathrm{Pr}_{p_0}\left[A_f \geq \alpha\right]$

[4]Viewed here formally as emergent weight/pre-activation structures enabling the outlier.

Using statistical mechanics based notation, we define the Hamiltonian $H_{p,\alpha}$ and fluctuating "partition function" $Z_{A_f}$

$$H_{p,\alpha}(h) = \frac{\alpha^2}{2\left\langle y, \tilde{K}_{L-1}, y \right\rangle} + \frac{1}{2} \sum_{l=1}^{L-1} \sum_{i=1}^{N_l} \left\langle h_i^l, \tilde{K}_{l-1}^{-1}, h_i^l \right\rangle, \tag{9}$$

$$\log Z_{A_f}(h) = \log \left\langle y, \tilde{K}_{L-1}, y \right\rangle + \sum_{l=1}^{L-1} \operatorname{Tr} \log \tilde{K}_{l-1}.$$

Consequently, $p_{A_f}(\alpha)$ takes the form $p_{A_f}(\alpha) = \int \mathcal{D}h \, \exp\left(-H_{p,\alpha}(h) - \log Z_{A_f}(h)\right)$, where the integrand defines an effective measure on the preactivation space for a given $\alpha$. We next approximate this measure per $\alpha$ by an analytically tractable variational estimate, $q$. We follow a similar convention here: for any $q, \alpha$, we define $q_\alpha(h) := Z_{q,\alpha}^{-1} e^{-H_{q,\alpha}(h)}$, requiring that the minimum value of $H_{q,\alpha}(h)$ be zero. For a Gaussian $q$, this reduces exactly to Eq. 9, with kernels $\tilde{K}_{l-1}$ that are independent of $h$. The variational computation follows by looking for $q_\alpha(h)$ that minimizes the KL divergence between the measure on $h$ which defines $p_{A_f}$, and $q$. The KL divergence can also be used in the estimation of $E(\alpha) := -\log(p_{A_f}(\alpha))$, following the Feynman–Bogoliubov inequality Kuzemsky (2015); Bogolubov and Jr (2009); Huber (1968). Here we provide a brief description – for the full derivation see App. C.2. By applying the Feynman–Bogoliubov inequality, we obtain an upper bound on $E(\alpha)$

$$E(\alpha) \approx \min_{q_\alpha}(\mathbb{E}_{h \sim q_\alpha}[\log(Z_{A_f}(h)/Z_{q,\alpha})] + \tilde{E}_q(\alpha)), \qquad \tilde{E}_q(\alpha) = \mathbb{E}_{h \sim q_\alpha}[H_{p,\alpha}(h) - H_{q,\alpha}(h)] \tag{10}$$

We argue in App. C.4 that for $\alpha \approx 1$, the log terms are subleading w.r.t. $\tilde{E}_q(\alpha)$. Defining $q_{*,\alpha}$ to be the measure that minimizes $\tilde{E}_q(\alpha)$, we obtain $E(\alpha) \approx \tilde{E}_{q_*}(\alpha)$. Next, we turn to estimating the variational energy $\tilde{E}_q(\alpha)$ Eq. (10) for $\alpha \sim 1$, omitting all $\alpha$ indices for brevity. In App. C.1 we simplify $p_{A_f}$, and show that the distribution in each layer depends only on the previous through a fluctuating non-linear operator. Next, we assume that this kernel is weakly fluctuating, and replace it with its expectation w.r.t. the variational distribution. This choice of approximation aligns with various works on deep non-linear networks, where layer-wise kernels are identified as the relevant and sufficient set of order parameters Rubin et al. (2025); Fischer et al. (2024); Seroussi et al. (2023c); Ringel et al. (2025). We further take a decoupled Gaussian variational ansatz so that $q(h) = \prod_{l=1}^{L-1} \prod_{i=1}^{N_l} q_{l,i}(h_i^l)$ where $q_{l,i}$ is Gaussian with mean $\mu_{l,i}$ and variance $Q_{l,i}$. As shown in App. C.3, the variational energy estimate is then given by

$$\tilde{E}_q \propto \sum_{l=1}^{L-1} \sum_{i=1}^{N_l} \underbrace{\left( \mathbb{E}_{h \sim \mathcal{N}(\mu_{l,i}, Q_{l,i})} \left[ \left\langle h, K_{l-1}^{-1} - Q_{l,i}^{-1}, h \right\rangle \right] + \left\langle \mu_{l,i}, Q_{l,i}^{-1}, \mu_{l,i} \right\rangle \right)}_{=: \Delta_{l,i}} + \underbrace{\left\langle y, K_{L-1}, y \right\rangle^{-1}}_{=: a_y}, \tag{11}$$

where we define $K_l = \mathbb{E}_{h \sim q}[\tilde{K}_l]$. Here, the $\Delta_{l,i}$ terms arise from the difference between the approximated kernel and the actual one, and the $a_y$ term results from enforcing an alignment $\alpha \approx 1$. Requiring that $q$ minimize $\tilde{E}_q$ and $\alpha \approx 1$, we estimate $\tilde{E}_q \propto E(\alpha \approx 1) \propto P_*$. Another interpretation of $\Delta_{l,i}$, discussed in App. In D, the excess weight is due to FL. This viewpoint is useful for FL patterns involving circuits, as the latter have a sharp imprint in weight-space. The above kernel viewpoint is, however, more general and can be used both for circuits and for more distributed learning patterns.

# 5 HEURISTICS FOR MANUAL COMPUTATION OF VARIATIONAL APPROXIMATION

## 5.1 FEATURE LEARNING PATTERNS

While the above variational approach allows a variety of candidate $q$'s, we focus on the previously mentioned set of feature-learning scenarios that have been extensively studied in the literature. Although this subset may appear restrictive, by varying behaviors among layers and between different neurons of the same layer, it already captures a wide range of phenomena. We then need to compare

the variational energy ($\tilde{E}_q$), as detailed in Sec. 4, for such combinations and select the minimizer. The optimal pattern is an indication of the FL that emerges in the network to enable strong alignment, as motivated in App. A.3. Concretely, per layer and neuron pre-activation ($h_i^l(x)$), we allow one of the following choices:

**(1) Gaussian Process (GP).** Here, $q_{l,i}$ is a Gaussian process (GP) so that $h_{l,i} \sim \mathcal{N}(0, K_{l-1})$ with $K_{l-1}$ defined as the expectation of the kernel defined in (8). This choice defines the "base model" of FL. For FCNs [5], it implies that the network propagates feature structure forward without altering latent features (see Sec. 5.2). When all layers and neurons follow this distribution, the network reduces to the neural network GP (NNGP) Neal (1996), where no FL occurs. Introducing any of the patterns below in a subset of neurons enables FL to emerge.

**(2) Gaussian Feature Learning (GFL).** In this scenario, pre-activations remain Gaussian with zero mean, but the covariance is modified relative to the GP scenario (1): the kernel of the previous layer is amplified by a factor $D$ in the direction of a specific feature (e.g., an eigenfunction of $K_{l-1}$) $\Phi_*^l$, one may also consider generalizations to several features). Thus, here too, the distribution is a GP but with a different covariance $Q_{l,i}$ given by

$$Q_{l,i}(x, x') = K_{l-1}(x, x') + D\langle \Phi_*^l, K_{l-1}, \Phi_*^l \rangle \Phi_*^l(x)\Phi_*^l(x'). \tag{12}$$

**(3) Specialization.** In this scenario, a given neuron specializes to a particular feature $\Phi_*^l$ with proportionality constant $\mu_{l,i}$. This pattern corresponds to a Gaussian distribution which is sharply peaked around a non-zero mean $\mu_{l,i}\Phi_*^l$ [6]. Explicitly, we define the distribution of the specialized neuron as

$$q_{l,i}(\langle h_i^l, \Phi_*^l \rangle) = \delta[\langle h_i^l, \Phi_*^l \rangle - \mu_{l,i}], \quad q_{l,i}(\langle h_i^l, \Phi_\perp^l \rangle) = \mathcal{N}(0, \langle \phi_\perp^l, K_{l-1}, \Phi_\perp^l \rangle). \tag{13}$$

## 5.2 LAYER-WISE FEATURE PROPAGATION

Since the variational energy of each layer depends on the kernel of the previous layer, an important element in our heuristic is understanding how the choice of pattern in a given layer affects the kernel and its spectrum in the subsequent layer. To this end, we define feature learning as any deviation from the baseline GP pattern (see Sec. 5.1), such as introducing a non-zero mean to the distribution (i.e., specialization) or altering its covariance structure (i.e., GFL). In our framework, a "feature" refers either to the mean $\mu_{l,i}$ of $q_{l,i}$ or to an eigenfunction of its covariance operator $Q_{l,i}(x, x')$.

We now outline several key claims concerning how features typically propagate between layers in FCNs. In this context, we consider a data measure that is i.i.d. Gaussian with zero mean and variance 1, not because it approximates the data well, but rather because it provides an unbiased baseline (see also Lavie and Ringel (2025)) for measuring function overlaps. Depending on the input, other choices can also be considered (e.g., permutation-symmetric measures over discrete tokens Lavie et al. (2024)). The following claims with their justifications should be understood as heuristic principles or rationalizations of empirically observed phenomena. Proving them in general or augmenting for different architectures is left for future work. For further details and empirical results, see App. C.5.

**Claim (i): Neuron specialization creates a spectral spike.** Assume that $M$ neurons in layer $l$ specialize on a single feature $\Phi_*^l(x)$, the subsequent kernel $K_l$ develops a new, dominant spectral feature corresponding to $\sigma(\Phi_*^l(x))$. The corresponding RKHS norm of this feature is amplified, scaling as $\mathcal{O}(N_l/M)$. **Justification:** When $M$ neurons specialize, the next layer's kernel is approximately $K_l(x, x') = A(x, x') + \frac{M}{N_l}\sigma(\Phi_*^l(x))\sigma(\Phi_*^l(x'))$, where $A$ is the contribution from the non-specialized neurons. Treating the specialization term as a rank-1 update, the Sherman-Morrison formula shows that its RKHS norm becomes $(R_A^{-1} + M/N)^{-1}$, where $R_A$ is the RKHS norm of $A$, which satisfies $R_A^{-1} \ll M/N$ in typical high-dimensional settings.

**Claim (ii): Amplified features in the pre-activation kernel create amplified higher-order features in the post-activation kernel.** If a feature $\Phi_*^l(x)$ in kernel $K_l$ has its eigenvalue enhanced by

---

[5]For CNNs, even in the lazy regime, deeper kernels have a different input scope and hence generate a new structure.

[6]Taking equilibrated networks and increasing the amount of data, specialization was shown in Rubin et al. (2024) to emerge as a first-order phase transition where the average of preactivations suddenly shifts to $\mu_{l,i}$. This behavior was further associated with Grokking, suggesting a potential specialization-grokking link.

a factor $D$ (i.e., $\lambda_* \to \lambda_* D$), then the corresponding $m$-th order power of this feature $(\Phi_*^l)^m(x)$ will have the bulk of its spectral decomposition, under the downstream kernel, shifted up by $D^m$, with similar effect on the inverse RKHS norm. **Justification:** A Taylor expansion of $K_{l+1}$ in terms of the eigenfunctions of $K_l$ shows that the term corresponding to $(\Phi_*^l)^m(x)$ will have a coefficient scaling with $(\lambda_* D)^m$. We argue that this term is difficult to span using other terms in this expansion, allowing us to treat it as a spectral spike and analyze it similarly to Claim (i). A numerical demonstration of this effect is shown in Fig. 5.

**Claim (iii): Lazy layers preserve the relative scale of features from the previous layer.** In the absence of FL, a properly normalized lazy layer approximately preserves the eigenspectrum of the previous kernel. If a feature $\Phi_*^l(x)$ has an eigenvalue $\lambda_*$ with respect to the pre-activation kernel given by $K_{l-1}$, its effective eigenvalue with respect to the post-activation kernel $K_l$ will also be proportional to $\lambda_*$. **Justification:** Follows from Claim (ii) taking $D = 1$.

---

Propagation rules for FCNs

(1) **Specialization**: Layer $l$ specialized $M$ neurons on $\Phi_*^l$. For any feature $\Phi$ satisfying $\langle \sigma(\Phi_*^l), \Phi \rangle \neq 0$, we approximate $\langle \Phi, K_l^{-1}, \Phi \rangle \propto \left[ \sum_{i \text{ sp.}} \frac{\mu_{i,l}^2}{N_l} \right]^{-1}$, where we sum over all specializing neurons.

(2) **GFL**: Layer $l$ amplified fluctuation along $\Phi_*^l$ by $D$ so that $\langle \Phi_*^l, K_l^{-1}, \Phi_*^l \rangle = (D\lambda_*)^{-1}$, where $\lambda_*$ is the GP value of the inner product. Then for any $m$ we have $\langle (\Phi_*^l)^m, K_l^{-1}, (\Phi_*^l)^m \rangle \propto (D\lambda_*)^{-m}$.

---

## 6 CONCRETE EXAMPLES

We now apply the heuristic principles of Sec. 4, 5.1, 5.2 to derive sample complexity bounds in a few examples. In App. E.1, we benchmark our method on two-layer FCNs and simple CNNs with non-overlapping patches. There we reproduce both the sample complexity exponent $P_* = d^{3/4}$ identified for CNNs in Ringel et al. (2025) and further predict that $P_* = d$ for two-layer FCNs studying a Hermite-3 target as well as the scaling of the number of specializing neurons with width (see Fig. 2). The latter is also a setting for which the prior's upper bound can be computed directly from $E(\alpha)$ using Large Deviation Theory, leading to a good match with experiment (Fig. 2 panel (a)).

Going to what we believe is beyond the current analytical state of the art, in App. E.3, we predict a $P_* = \sqrt{L}$, where $L$ is the context length, of a softmax attention layer learning a cubic target (see Fig. 3). Another such instance, discussed in detail below, is that of a 3-layer non-linear network learning a non-linear target. In App. A.2 we show that this approach can be extended to classification tasks as well, on non-Gaussian data. In App. E.2 we apply our heuristics to a concrete setting, of a two-layer ReLU network trained on a parity task. We find there as well are able to predict the emergent feature pattern, which qualitatively differs from erf networks. Rather than identifying the scaling number of specializing neurons, we find that there is a finite number of neurons and we are able to predict their scale.

### 6.1 THE THREE-LAYER NETWORK

Here we consider three-layer FCNs given by $f(x) = \sum_{i=1}^{N_2} a_i \sigma\left( \sum_{j=1}^{N_1} w_{ij}^{(2)} \sigma(w_j^{(1)} \cdot x) \right)$, where $x \in \mathbb{R}^d$ is drawn from $\mathcal{N}(0, I_d)$. We train these networks on a polynomial target of degree $m$ given by $y(x) = He_m(w_* \cdot x)$ where $He_m$ is the $m$-th probabilist Hermite polynomial, which is the standard polynomial choice under our choice of data measure, and $w_* \in \mathbb{R}^S$ is some normalized vector. The networks are trained via Langevin dynamics Welling and Teh (2011b), with ridge parameter $\kappa$, quadratic weight decay, and standard scaling. For an extension to mean-field scaling, see App. E.1.1.

As a starting point for our analysis, consider the simplest pattern ($q$"="**GP-GP**), having two GP/lazy layers where taking an $l$'th layer to be lazy means $Q_{l,i} = K_{l-1}$ and $\mu_{l,i} = 0$. Following the choice of pattern, our goal is to estimate the scaling of $\tilde{E}_q$. Examining Eq. (11), we find that the $\Delta_{l=1,2,i}$

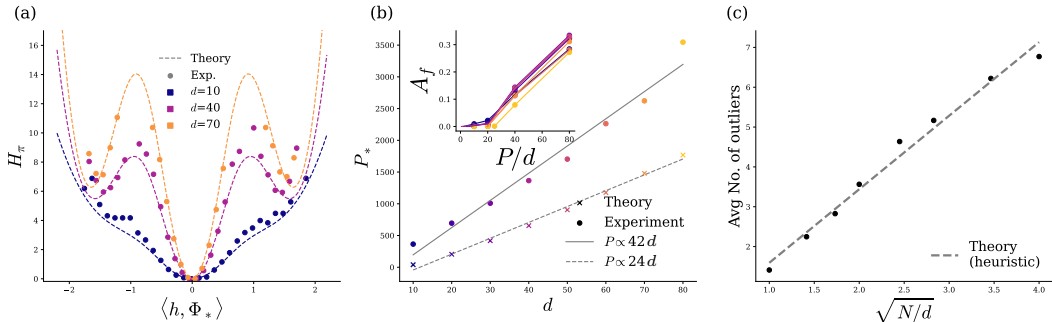

Figure 2: Numerical and experimental results for a two-layer erf network trained on the normalized third Hermite polynomial ($m = 3$). In panel (a) we compare the experimental results and exact theoretical predictions (computed utilizing LDT, see App. B) for the distribution of the alignment of the hidden layer pre-activation with the linear feature. Here we follow the same notation as in (9), so that $H_\pi$ is the negative log posterior of the preactivations up to an additive constant that enforces zero minimum. We also find the pre-activation distribution corresponds to $q(h)$ for $q \sim$ M-Sp, as predicted by our heuristic approach. Panel (b) compares theoretical and experimental predictions for $P_*$, defined as alignment $\alpha > 0.1$ (inset shows alignment as a function of sample size). Both theoretical and experimental results agree on $P_* \propto d$. In (c), we increase $N$ and keep $P$ and $d$ fixed, and plot the number of specialized neurons in the hidden layer. In agreement with our heuristic predictions, the number of neurons increases linearly with $\sqrt{N/d}$.

contributions all cancel by our above choice of $Q$'s and $\mu$'s. The only non-zero contribution thus comes from the final layer and is given by the inverse of $\langle y, K_2, y \rangle$. Because of lazy learning, $K_2(x, x')$ is a standard dot product FCN kernel which can be expanded as $\sum_{n=1}^{\infty} a_n (x \cdot x'/d)^n$, with $a_n = O(1)$ w.r.t. $N_{1,2}, d, P$. It can then be shown that $\langle y, K_2, y \rangle = O(d^{-m})$. Leading to $\tilde{E}_{GP-GP} = a_y \propto d^m$.

Next, we consider a FL pattern wherein the first layer is GP distributed but the second has FL ($q$"="**GP-Sp.**). Specifically, for the $l = 1$ layer, we take $Q_{1,i} = K_0; \mu_{1,i} = 0$ thereby nullifying again $\Delta_{1,0}$ in Eq. 11 for $\tilde{E}_q$. For the second layer, we assume $M_2$ specializing neurons (e.g., $i = 1..M_2$) which fluctuate around the linear feature ($w_* \cdot x$), while others are lazy, namely $Q_{2,i>M_2} = K_1, \mu_{2,i>M_2} = 0$ and $Q_{2,i=1..M_2} = K_1, \mu_{2,i=1..M_2}(x) = (w_* \cdot x)$. Examining $\sum_{i=1}^{N_2} \Delta_{2,i}$ in Eq. 11, we get zero contributions from $\Delta_{2,i>M_2} = 0$ and $M_2 \langle (w_* \cdot x), K_1^{-1}, (w_* \cdot x) \rangle$ from $i = 1..M_2$. As $K_1$ is again a simple FCN dot product kernel with no FL effects, normalized linear functions such as $w_* \cdot x$ have an $O(d)$ RKHS norm. We thus obtain an overall contribution to $\tilde{E}_q$ from the $l = 2$ layer equal to $M_2 d$. Finally, we need to estimate $a_y = \langle y|K_2|y \rangle^{-1}$. Note that $K_2$ is not a standard FCN kernel anymore, since $Q_2$, which contains target information, is used in its definition as the expectation of the kernel appearing in (Eq. 8). According to our feature propagation rule (i), with $\Phi_* = (w_* \cdot x)$, we have $a_y = N_2/M_2$. Given $M_2$, we thus obtain a variational energy of $M_2 d + N_2/M_2$. We next need to minimize over free parameters, namely $M_2$ leading to $M_2 = \sqrt{N_2/d}$ and finally $\tilde{E}_{\mathbf{GP-Sp.}} = \sqrt{N_2 d}$. Provided $N_2$ scales less than $d^{2m-1}$ ($N_2 = o(d^{2m-1})$), this pattern is favorable to $\mathbf{GP - GP}$.

Finally, we consider what turns out to be the favorable pattern consisting of $M_1$ neurons specializing on $(w_* \cdot x)$ in the first layer and all neurons in the second layer specializing $He_m(w_* \cdot x)$, with a small proportionality constant $\mu_{2,i} = \pm\sqrt{\beta/N_2}$ ($q$"="**Sp.-Mag.**). We refer to the second-layer pattern as magnetization. Following straightforward adaptation previous argument to this pattern, the variational energy for this pattern as well as others, for $m = 3$, can be found in Table 1.

In the non-GP $q$ patterns, we obtain the same scaling of $\tilde{E}_q$ in the proportional limit ($N_1 \propto N_2 \propto d$), namely, $P_*/\kappa \propto d$. This observation is validated experimentally in Fig. 3, where the transition to non-zero alignment becomes sharper in the thermodynamic limit ($d \to \infty$). However, the mechanism by which this scaling is realized changes. In the specialization-magnetization pattern the sample complexity scales with $N_1^{1/3}$, therefore, it increases with $N_1$. However, under the GP-

| Feature Pattern | $\Delta_1$ | $\Delta_2$ | $a_y$ | Minimizing Parameters | $\tilde{E}$ |
|---|---|---|---|---|---|
| **GP-GP** | 0 | 0 | $d^3$ | — | $d^3$ |
| **GP-Sp.** | 0 | $M_2 d$ | $N_2/M_2$ | $M_2 = \sqrt{N_2/d}$ | $\sqrt{N_2 d}$ |
| **Sp.-Mag.** | $M_1 d$ | $N_1\beta/M_1$ | $N_2/\beta$ | $\beta = \left(N_2^2/N_1 d\right)^{1/3}, \quad M_1 = \left(N_2 N_1/d^2\right)^{1/3}$ | $(N_1 N_2 d)^{1/3}$ |

Table 1: Variational energy $\tilde{E}$ for different choices of feature-learning patterns in a three-layer FCN trained on $y(x) = \mathrm{He}_3(w_* \cdot x)$. The patterns shown here are (first/second layer): GP-GP, GP-Specialization, and Specialization-Magnetization. For each pattern, the components of the variational energy ($\Delta_1, \Delta_2, a_y$) together with the corresponding minimizing parameters are shown. We comment that the GP-GP pattern is favorable only for $d > N_2^5$, and otherwise FL will emerge.

specialization pattern, sample complexity does not scale with $N_1$, making this pattern preferable. This prediction is in line with experimental results (see Figs. 8 and 3 panel (c)) where increasing $N_1$ causes the described change in FL patterns. Our prediction also accurately determines the scaling of the number of specializing neurons with $N_1$.

## 6.2 SOFTMAX ATTENTION

Here, we consider an attention block of the form

$$f(X) = \frac{1}{\sqrt{L}} \sum_{h=1}^{H} \sum_{a,b=1}^{L} @_{ab;h}(X)(w_h \cdot x^b), \quad @_{ab;h}(X) = e^{[x^a]^\top A_h x^b} \Big( \sum_{c=1}^{L} e^{[x^a]^\top A_h x^c} \Big)^{-1}, \quad (14)$$

where $X \in \mathbb{R}^{L \times d}$, $A \in \mathbb{R}^{d \times d}$, and $x^a \in \mathbb{R}^d$ is the $a$-th row of $X$, and $w_h \in \mathbb{R}^d$. Our prior on network weights is $\prod_{h=1}^{H} \mathcal{N}(0, I_{d^2}/d^2; A_h) \mathcal{N}(0, I_d/(dH); w_h)$. The only dependence on the context length $L$ arises from the pre-factor $1/\sqrt{L}$, which ensures that for $X_i^a \sim \mathcal{N}(0,1)$, we have $f(X) = \mathcal{O}(1)$. The target function is given by $y(X) = \sum_{a,b} \frac{1}{\sqrt{L(L-1)}} x_1^a x_2^b x_3^b$, also normalized to be $\mathcal{O}(1)$. Following our approach, we propose two patterns for this architecture: GP (or lazy learning) and specialization (where we take $A_h \sim \mathcal{N}(\mu\sigma_x \otimes I_{(d-2)}, I_{d^2})$ for $\sigma_x = [1,0;0,1]$ and optimize over $\mu$). As detailed in E.3, the variational energy scales as $Ld^3$ for the GP pattern and as $\sqrt{HLd^3}$ for the specialization pattern, the latter thus being favorable for $H$ scaling less than $\sqrt{Ld^3}$. As shown in Fig. 3, this scaling of $P$ with $L$ and $d$ indeed matches the dependence of the sample complexity on $L$.

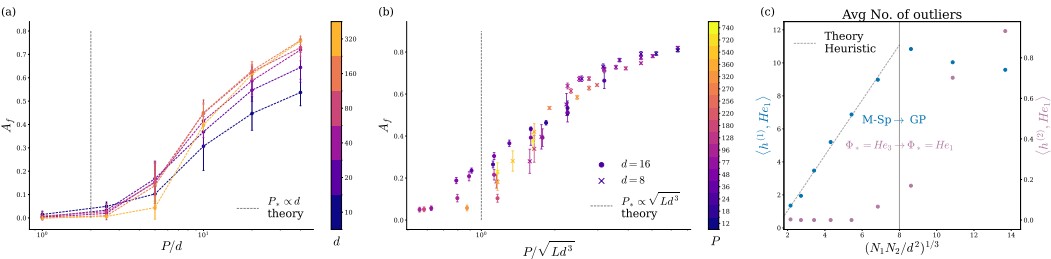

Figure 3: **Sample complexity:** Heuristic predictions accurately capture sample complexity in both three-layer erf FCNs and softmax attention heads, as well as feature learning scaling. Panels (a) and (b) track network alignment against the ratio between the sample size, $P$, and the predicted sample complexity- $P/d$ for FCNs in panel (a), and $P/\sqrt{Ld^3}$ for attention heads in panel (b). In both cases, alignment collapses onto a single curve, confirming the sample complexity predictions. See Fig. 9 for MSE comparison and Apps. E.3, F.1 for further details. **FL patterns:** Panel (c) tracks linearly specialized neurons in the first (blue) and second (purple) layers against $N_1$ (with fixed $P, d$ and $N_2$). The number of first-layer specializing neurons initially follows the predicted $(N_1/d)^{(1/3)}$ scaling until the transition where second-layer neurons specialize on the linear feature, and the first layer approaches the GP distribution.

## 7 DISCUSSION

This paper presents a novel methodology for analyzing the scaling behavior of sample complexity, through which we can also understand how distinct learning mechanisms emerge. Its strength lies in abstracting away from fine-grained details to isolate the core principles at play. We hope such a strategy would remove barriers and expedite connections between mechanistic interpretability and first-principles scientific approaches.

**Limitations.** Notwithstanding, several avenues for improvement remain. In particular, quantifying feature propagation in more general CNNs and transformers, and addressing multi-feature interactions as in the context of superpositionElhage et al. (2022). It would also be desirable to extend our heuristic to dynamics of learning, potentially drawing insights from previous work relating equilibrium and dynamical phenomena Power et al. (2022); Rubin et al. (2024); Bahri et al. (2024); Nam et al. (2024). Since Bayesian convergence times can be slow, correctly predicting the emergence of FL in early stages of training may also be highly advantageous. Finally, in some cases, such as under mean-field scaling or vanishing ridge, overfitting effects can emerge, leaving our lower bound vacuous. Extending our approach to patterns that align only on the training set and incorporating effective ridge ideas Canatar et al. (2021a) is thus desirable.

## ACKNOWLEDGMENTS

Funded by the the Israeli Science Foundation - 374/23.

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
