MITIGATING THE CURSE OF DETAIL: SUPPLEMENTARY MATERIAL

## A  UPPER BOUNDS ON POSTERIOR ALIGNMENT

In this appendix, we derive bounds on the posterior probability of alignment between the network output and the target function. Our goal is to identify the scaling of the minimal sample size $P_*$ as a function of the input dimension, $d$, required for learning. The derivation proceeds by first relating posterior probabilities of trained network outputs to the prior. While the prior probability is estimated heuristically in C, we provide here a rigorous method for computing this bound using large-deviation techniques.

### A.1  THE POSTERIOR BOUND

Let $f_\Theta$ be the output of a network with a given set of trainable weights $\Theta$. Following standard derivations Neal (1996), the posterior distribution of possible outputs of such a network, conditioned on training on data $X \in \mathbb{R}^{P \times d}$ with respect to target $y : \mathbb{R}^d \to \mathbb{R}$ (which we assume is normalized for simplicity), with MSE loss and ridge parameter $\kappa$, is given by

$$\pi(f) = \frac{1}{Z} \exp\left(-\frac{|f(X) - y(X)|^2}{2\kappa}\right) p_0(f), \tag{15}$$

where $f$ and $y$ are applied row-wise to $X$, $|\cdot|$ is the $L_2$ norm on the training set, $p_0(f)$ is the prior distribution, defined by

$$p_0(f) = \int d\Theta\, p(\Theta)\, \delta[f - f_\Theta], \tag{16}$$

$p(\Theta)$ is the prior weight distribution, and $\delta[f - f_\Theta]$ is the functional delta function Ringel et al. (2025). We are interested in an upper bound on the posterior probability of achieving alignment $\geq \alpha$, where we define alignment by $A_f := \langle f, y \rangle = \int d\mu_x f(x) y(x)$. The prior alignment density is given by

$$p_{A_f}(\alpha) = \int d\Theta\, p(\Theta)\, \delta(A_{f_\Theta} - \alpha). \tag{17}$$

The prior and posterior probabilities of having alignment over some threshold $\alpha$ are given by $\mathrm{Pr}_{p_0}[A_f \geq \alpha]$ and $\mathrm{Pr}_\pi[A_f \geq \alpha]$, respectively. Since the loss is positive, we have $\mathrm{Pr}_\pi[A_f \geq \alpha] \leq \frac{1}{Z}\mathrm{Pr}_{p_0}[A_f \geq \alpha]$. We proceed by obtaining a lower bound on the partition function $Z$ – the normalization constant of the posterior distribution – given by

$$Z = \mathbb{E}_\Theta\left[e^{-\frac{|f_\Theta(X) - y(X)|^2}{2\kappa}}\right] := \int d\Theta\, p(\Theta)\, e^{-\frac{|f_\Theta(X) - y(X)|^2}{2\kappa}}. \tag{18}$$

By Jensen's inequality, we obtain

$$\exp\left(-\frac{\mathbb{E}_\Theta\left[|f_\Theta(X) - y(X)|^2\right]}{2\kappa}\right) < Z. \tag{19}$$

Since $f_\Theta \sim \mathcal{N}(0, K)$, for the NNGP kernel $K$ on the training data Cho and Saul (2009); Neal (1996), it follows that $\mathbb{E}_\Theta\left[|f_\Theta(X) - y(X)|^2\right] = \mathrm{Tr}(K) + |y(X)|^2$. For our choice of normalization, both $\mathrm{Tr}(K)$ and $|y(X)|^2$ scale with $P$. Thus, up to an $\mathcal{O}(1)$ factor $k = P^{-1} \sum_{\nu=1}^P \mathbb{E}_{p_0}[(f(x_\nu) - y(x_\nu))^2]$, we obtain $\mathbb{E}_\Theta\left[|f_\Theta(X) - y(X)|^2\right] = Pk$. Substituting this in the posterior upper bound, we obtain

$$\mathrm{Pr}_\pi[A_f \geq \alpha] < \exp\left(\frac{Pk}{2\kappa}\right)\mathrm{Pr}_{p_0}[A_f \geq \alpha], \tag{20}$$

or, equivalently,

$$\log\left(\Pr_\pi\left[A_f \geq \alpha\right]\right) < Pk/\left(2\kappa\right) + \log\left(\Pr_{p_0}\left[A_f \geq \alpha\right]\right). \tag{21}$$

The r.h.s. is thermodynamically large in magnitude, crossing zero only briefly, since large negative values imply a vanishingly small probability of strong alignment, the zero crossing marks the threshold for learning. The minimal sample size necessary for learning is then

$$P_* = -\frac{2\kappa}{k}\log\Pr_{p_0}\left[A_f \geq \alpha\right]. \tag{22}$$

We comment that $P_*$ provides an *unlearnability* bound, implying that learning cannot occur with less than $P_*$ . However, we find that $P_*$ is indeed indicative of the start of learning in the cases considered in this work. In certain settings, overfitting effects cause this bound to underestimate the true sample complexity, an effect that requires further study. In the following section, we compute the prior probability of alignment on the RHS using the Chernoff bound and large deviations theory, yielding a better estimation for $P_*$.

## A.2 CLASSIFICATION

In the case of a classification problem, we assume the output $f(x) \in \mathbb{R}^{d_o}$, where $d_o$ denotes the number of categories. The dataset is given in the form of a hot-one encoding $\mathcal{D} = \{(x_\mu, y_\mu)\}_{\mu=1}^{P}$, where $y_\mu \in \mathbb{R}^{d_o}$ and $(y_\mu)_i = \delta_{c(x)i}$ for the classification target $c(x)$ taking values in $\{1, \ldots, d_o\}$. The cross-entropy loss for this setting is

$$\mathcal{L}(f; \mathcal{D}) = -\sum_{\mu=1}^{P}\left[\log\left(\frac{e^{-f_{c(x_\mu)}(x_\mu)}}{\sum_{j=1}^{d_o}e^{-f_j(x_\mu)}}\right)\right]$$

$$= \sum_{\mu=1}^{P}\left[f_{c(x_\mu)}(x_\mu) + \log\left(\sum_{j=1}^{d_o}e^{-f_j(x_\mu)}\right)\right] \geq 0$$

The derivation

$$\pi\left(f\right) = \frac{1}{Z}\exp\left(-\frac{\mathcal{L}(f; \mathcal{D})}{2\kappa}\right)p_0\left(f\right), \tag{23}$$

follows that of the MSE loss. We similarly have $\Pr_\pi\left[A_f \geq \alpha\right] \leq \frac{1}{Z}\Pr_{p_0}\left[A_f \geq \alpha\right]$, since the cross-entropy loss is also positive. The use of Jensen's inequality does not depend on $\mathcal{L}(f; \mathcal{D})$, so we only need to compute $\mathbb{E}_\Theta\left[\mathcal{L}(f; \mathcal{D})\right]$.

By Cho and Saul (2009); Neal (1996), we have $f_j \sim \mathcal{GP}\left(0, K_j\right)$ for the NNGP kernel $K_j$. Since we only care about the values of $f$ at $\mathcal{D}$, we consider $f_j \sim \mathcal{N}(0, K_j)$, $K_j \in \mathbb{R}^{P \times P}$, $(K_j)_{\mu\nu} = K_j(x_\mu, x_\nu)$, where we overloaded the notation $f_j$ and $K_j$ for simplicity.

$$\mathbb{E}_\Theta\left[\mathcal{L}(f; \mathcal{D})\right] = \mathbb{E}_{f \sim \prod_{j=1}^{d_o}\mathcal{N}(0, K_j)}\left[\mathcal{L}(f; \mathcal{D})\right]$$

$$= \mathbb{E}_{f \sim \prod_{j=1}^{d_o}\mathcal{N}(0, K_j)}\left[\sum_{\mu=1}^{P}\left[f_{c(x_\mu)}(x_\mu) + \log\left(\sum_{j=1}^{d_o}e^{-f_j(x_\mu)}\right)\right]\right]$$

$$= \sum_{\mu=1}^{P}\mathbb{E}_{f \sim \prod_{j=1}^{d_o}\mathcal{N}(0, K_j)}\left[f_{c(x_\mu)}(x_\mu) + \log\left(\sum_{j=1}^{d_o}e^{-f_j(x_\mu)}\right)\right]$$

Since each summand depends on $x_\mu$ alone, we can replace $f \sim \prod_{j=1}^{d_o} \mathcal{N}(0, K_j)$ by $f_\mu \sim \mathcal{N}(0, \Sigma_\mu)$ where $(\Sigma_\mu)_{ij} = \delta_{ij}\sigma_{\mu,j}^2$, $i, j \in \{1, \ldots, d_o\}$, $\sigma_{\mu,j}^2 = K_j(x_\mu, x_\mu)$. We will also denote $c_\mu := c(x_\mu)$.

$$\mathbb{E}_\Theta\left[\mathcal{L}(f; \mathcal{D})\right] = \sum_{\mu=1}^{P} \mathbb{E}_{f_\mu \sim \mathcal{N}(0, \Sigma_\mu)}\left[(f_\mu)_{c_\mu} + \log\left(\sum_{j=1}^{d_o} e^{-(f_\mu)_j}\right)\right]$$

$$= \sum_{\mu=1}^{P} \mathbb{E}_{f_\mu \sim \mathcal{N}(0, \Sigma_\mu)}\left[\log\left(\sum_{j=1}^{d_o} e^{-(f_\mu)_j}\right)\right]$$

Let $\psi(f_\mu) := \log\left(\sum_{j=1}^{d_o} e^{-(f_\mu)_j}\right)$, so we end up with

$$\mathbb{E}_\Theta\left[\mathcal{L}(f; \mathcal{D})\right] = \sum_{\mu=1}^{P} \mathbb{E}_{f_\mu \sim \mathcal{N}(0, \Sigma_\mu)}\left[\psi(f_\mu)\right]$$

Let us concentrate on a single summand and drop $\mu$ from our notation for the moment. By Wick's Theorem, we have

$$\mathbb{E}_{f \sim \mathcal{N}(0, \Sigma)}\left[\psi(f)\right] = e^{\frac{1}{2}\sum_{j=1}^{d_o}\sigma_j^2 \frac{\partial_j^2}{\partial f_j^2}}\ \psi(f)|_{f=0}$$

$$= \psi(0) + \frac{1}{2}\sum_{j=1}^{d_o}\sigma_j^2 \frac{\partial_j^2}{\partial f_j^2}\ \psi(f)|_{f=0} + \text{H.O.T}$$

$$= \log(d_o) + \frac{d_o - 1}{d_o^2}\sum_{j=1}^{d_o}\sigma_j^2 + \text{H.O.T}$$

Bringing the sum over datapoints back and assuming $\text{Tr}(K_j)$ scales as $P$, we have

$$\mathbb{E}_{f \sim \prod_{j=1}^{d_o} \mathcal{N}(0, K_j)}\left[\mathcal{L}(f; \mathcal{D})\right] = P\log(d_o) + \underbrace{\frac{1}{d_o}\sum_{j=1}^{d_o}\text{Tr}(K_j)}_{\mathcal{O}(P)} + \mathcal{O}\left(\frac{P}{d_o}\right)$$

where the subleading correction $\mathcal{O}(P/d_o)$ comes from an $\mathcal{O}(1/d_o)$ contribution of the $2n$-th derivative for $n \geq 2$ combined with summation over $d_o$ classes and $P$ stems from the summation over datapoints. Thus, for a classification task, we obtain

$$\Pr_\pi[A_f \geq \alpha] < \exp\left(\frac{P\log(d_o)}{2\kappa}\right)\Pr_{p_0}[A_f \geq \alpha], \tag{24}$$

Consequently, Eq. 22 applies with the necessary adjustment, replacing $k$ with $\log(d_o)$.

Our use of Wick's Theorem to compute the disorder–averaged log-partition function over classes is closely related to the annealed and quenched free-energy calculations that appear in the statistical-physics literature on disordered systems. In that language, for fixed data $\mathcal{D}$ and random parameters $\Theta$, the quantity $-\mathcal{L}(f_\Theta; \mathcal{D})$ plays the role of a random energy, and $-\mathbb{E}_\Theta[\mathcal{L}(f_\Theta; \mathcal{D})]$ is the quenched free energy of a disordered system with $d_o$ states. Our leading term $P\log(d_o)$ is then the purely entropic contribution coming from $d_o$ equiprobable states per datapoint, in direct analogy with the $N\log(2)$ term in Derrida's random–energy model Derrida (1980) for a system with $2^N$ configurations. In the spin-glass literature, one often replaces the quenched

average $\mathbb{E}[\log(Z)]$ by the annealed approximation $\log\mathbb{E}[Z]$, which follows from Jensen's inequality for $\log$ and is accurate in the replica-symmetric/high-temperature phase. Our calculation stays on the quenched side and is elementary (Gaussian integrals plus Wick's theorem), but the resulting scaling structure – an entropic $\log(d_o)$ term with subleading corrections controlled by the covariance of the random energies – is entirely consistent with this broader statistical-mechanics viewpoint.

### A.3 THE CHERNOFF UPPER BOUND

We refine the posterior bound using the Chernoff inequality. This allows us to estimate the probability of achieving alignment $\geq \alpha$ under the prior distribution, leading to an explicit expression for the corresponding sample complexity (22). From the Chernhoff bound, we get

$$\Pr_{p_0}\left[A_f \geq \alpha\right] \leq \inf_{t>0} e^{-t\alpha}\mathbb{E}_\Theta\left[e^{tA_f}\right] =: e^{-E(\alpha)}. \tag{25}$$

Substituting in (22), we obtain a new estimate

$$P_* = \frac{2\kappa}{k}E(\alpha). \tag{26}$$

Our aim is to better estimate $E(\alpha)$. Following standard complex analysis derivation, we have

$$-\log(\Pr_{p_0}\left[A_f \geq \alpha\right]) \geq E(\alpha) := -\log(\inf_{t>0}\mathbb{E}_\Theta\left[\exp\left(t\left(A_{f_\Theta} - \alpha\right)\right)\right])$$
$$= -\log(\underset{t\in\mathbb{C}}{\text{ext}}\,\mathbb{E}_\Theta\left[\exp\left(it\left(A_{f_\Theta} - \alpha\right)\right)\right]), \tag{27}$$

where $\text{ext}$ denotes the extremum with respect to a complex $t$. Since we are in the rare event regime, we can apply a saddle point approximation to the Fourier transform of $p_{A_f}(\alpha)$

$$p_{A_f}(\alpha) = \frac{1}{2\pi}\int dt\mathbb{E}_\Theta\left[\exp\left(it\left(A_{f_\Theta} - \alpha\right)\right)\right] \approx \underset{t\in\mathbb{C}}{\text{ext}}\,\mathbb{E}_\Theta\left[\exp\left(it\left(A_{f_\Theta} - \alpha\right)\right)\right]. \tag{28}$$

Thus, (25) can be reduced to

$$-\log\Pr_{p_0}\left[A_f \geq \alpha\right] \geq -\log p_{A_f}(\alpha) \approx E(\alpha). \tag{29}$$

This definition of the upper bound allows for furhter simplifications.

## B AN EXPLICIT EQUATION FOR THE LDT BOUND IN FULLY CONNECTED NETWORKS

To make our LDT analysis concrete, we now focus on FCNs. This can be trivially adapted to other architectures, however, such as CNNs and transformers. Here, we provide an explicit expression for the minimal $t_*$ in Eq. 25. The resulting formula involves high-dimensional, non-Gaussian integrals and is therefore highly computationally intensive. We also work out the two-layer case explicitly, where we show that the solution can be reduced to one-dimensional integrals.

### B.1 SETUP

We begin by specifying the architecture and notation used throughout this section. Focusing on FCNs of depth $L > 1$, we define the forward pass and parameterization explicitly:

$$f(x) = \sum_{i=1}^{N_{L-1}} w_i^L \sigma\left(h_i^{L-1}(x)\right)$$

$$h_i^l(x) = \sum_{j=1}^{N_{l-1}} W_{ij}^l \sigma\left(h_j^{l-1}(x)\right) \quad l = 2, ..., L-1, \quad i = 1, ..., N_l$$

$$h_i^1(x) = [W^1 x]_i, \quad i = 1, ..., N_1$$

We train the network using Langevin dynamics Welling and Teh (2011a) with MSE loss on a target function $y$. The quadratic weight decay for each layer $l$ is set to $\kappa N_{l-1}/\sigma_l^2$ where $\kappa$ is the ridge parameter (corresponding to the Langevin dynamics temperature, $T$ Ringel et al. (2025)) and $N_0 = d$ is the input dimension. We define $\sigma_L^2, \kappa \sim \mathcal{O}(1/\chi)$, where we refer to $\chi$ as the mean-field (MF) scale. This choice of weight decay results in a Gaussian prior distribution for the weights given by $W_{i,j}^l \sim \mathcal{N}(0, \sigma_l^2/N_{l-1})$ for $i = 1, .., N_l$, $j = 1, ..., N_{l-1}$.

### B.2 DERIVATION

In the following, we will derive an explicit equation for $t_*$ that minimizes the upper bound for $\Pr_{p_0}[A_f \geq \alpha]$ in (25). We will use $\Theta^l$ to denote the weights up to (and including) layer $l$, in particular, $\Theta^L \equiv \Theta$. Utilizing this notation, we can separate the readout weights,

$$\mathbb{E}_\Theta\left[e^{t(A_f - \alpha)}\right] \propto$$

$$e^{-t\alpha} \int \prod_{i=1}^{N_{L-1}} dw_i^L d\Theta^{L-1} p\left(\Theta^{L-1}\right) \exp\left(-\frac{N_{L-1}}{2\sigma_L^2}\left(w_i^L\right)^2 + t\sum_{i=1}^{N_L} w_i^L \left\langle \sigma\left(h_i^{L-1}(x)\right), y(x)\right\rangle\right). \quad (30)$$

We integrate out the weights of the final layer to get

$$\mathbb{E}_\Theta\left[e^{t(A_f - \alpha)}\right] \propto e^{-t\alpha} \int d\Theta^{L-1} p\left(\Theta^{L-1}\right) \exp\left(\frac{\sigma_L^2}{2N_{L-1}} t^2 \underbrace{\sum_{i=1}^{N_{L-1}} \left\langle \sigma\left(h_i^{L-1}(x)\right), y(x)\right\rangle^2}_{A_\sigma^2(\Theta^{L-1})}\right). \quad (31)$$

If $\alpha$ is such that it is smaller than the upper bound of $A_f$, which occurs naturally for unbounded $f$ as in our case where the readout weights are unbounded, then $t_* \in (0, \infty)$. Thus, $t_*$ is determined by minimizing

with respect to $t$, and we obtain

$$0 = \left. \frac{\partial \mathbb{E}_\Theta \left[ e^{t(A_f - \alpha)} \right]}{\partial t} \right|_{t=t_*}$$

$$= -\alpha e^{-t\alpha} \int d\Theta^{L-1} p\left(\Theta^{L-1}\right) \exp\left( \frac{\sigma_L^2}{2N_{L-1}} \sum_{i=1}^{N_{L-1}} t^2 A_{\sigma,i}^2 \left(\Theta^{L-1}\right) \right)$$

$$+ \left. \sum_{i=1}^{N_{L-1}} \frac{\sigma_L^2}{N_{L-1}} t e^{-t\alpha} \int d\Theta^{L-1} p\left(\Theta^{L-1}\right) A_{\sigma,i}^2 \left(\Theta^{L-1}\right) \exp\left( \frac{\sigma_L^2}{2N_{L-1}} \sum_{i=1}^{N_{L-1}} t^2 A_{\sigma,i}^2 \left(\Theta^{L-1}\right) \right) \right|_{t=t_*},$$

where we have $A_{\sigma,i}^2 \left(\Theta^{L-1}\right) := \left\langle \sigma\left(h_i^{L-1}\right), y\right\rangle^2$. Rearranging, we obtain an implicit equation for $t_*$

$$t_* = \frac{\alpha N_L \int d\Theta^{L-1} p\left(\Theta^{L-1}\right) \exp\left( \frac{\sigma_L^2}{2N_{L-1}} t_*^2 A_\sigma^2 \left(\Theta^{L-1}\right) \right)}{\sigma_L^2 \int d\Theta^{L-1} p\left(\Theta^{L-1}\right) A_\sigma^2 \left(\Theta^{L-1}\right) \exp\left( \frac{\sigma_L^2}{2N_{L-1}} t_*^2 A_\sigma^2 \left(\Theta^{L-1}\right) \right)}, \tag{32}$$

which takes a simpler form, using the notation $\mathbb{E}_{\Theta^{L-1}}\left[(\cdots)\right] = \int d\Theta^{L-1} p\left(\Theta^{L-1}\right)(\cdots)$,

$$t_* = \frac{\alpha N_{L-1} \mathbb{E}_{\Theta^{L-1}} \left[ \exp\left( \frac{\sigma_L^2}{2N_{L-1}} t_*^2 A_\sigma^2 \left(\Theta^{L-1}\right) \right) \right]}{\sigma_L^2 \mathbb{E}_{\Theta^{L-1}} \left[ A_\sigma^2 \left(\Theta^{L-1}\right) \exp\left( \frac{\sigma_L^2}{2N_{L-1}} t_*^2 A_\sigma^2 \left(\Theta^{L-1}\right) \right) \right]}. \tag{33}$$

### B.3 EXACT LDT SADDLE-POINT COMPUTATION FOR A TWO-LAYER NETWORK

To illustrate the general framework in a more tractable setting, we analyze a simple two-layer FCN. In this case, the derivation simplifies substantially, allowing us to obtain explicit analytical expressions assuming $x \sim \mathcal{N}(0, I_d)$. Concretely, consider the setting

$$y(x) = \hat{H}e_3\left(w_* \cdot x\right), \quad \sigma = \mathrm{erf}, \quad L = 2$$

where $w_* \in \mathbb{R}^d$ is some normalized vector, and $\hat{H}e_3$ is the normalized third probabilist Hermite polynomial. Since this is a simple two-layer network, we need only consider the weights of the hidden layer themselves, and not the pre-activations. For brevity, we also drop layer indexing ($W \equiv W^1 = \Theta^1$, $N \equiv N_1$, $d = N_0$), since there is only one relevant layer, and take $\sigma_1^2, \sigma_2^2 = 1$, in particular, we assume standard scaling. Following (33), we obtain

$$t_* = \frac{\alpha N \mathbb{E}_W \left[ \exp\left( \frac{1}{2N} t_*^2 A_\sigma^2 (W) \right) \right]}{\mathbb{E}_W \left[ A_\sigma^2 (W) \exp\left( \frac{1}{2N} t_*^2 A_\sigma^2 (W) \right) \right]}$$

where $A_\sigma^2(W) = \sum_{i=1}^N \left\langle \sigma\left(w_i \cdot x\right), y(x)\right\rangle^2$, using $w_i \in \mathbb{R}^d$ for the $i$-th row of $W$. Since $W$ is i.i.d. in the different neuron indices, the above equation becomes

$$t_* = \frac{\alpha \int d^d w \exp\left( -\frac{d}{2} |w|^2 + \frac{1}{2N} t_*^2 A_\sigma^2 (w) \right)}{\int d^d w A_\sigma^2 (w) \exp\left( -\frac{d}{2} |w|^2 + \frac{1}{2N} t_*^2 A_\sigma^2 (w) \right)} \tag{34}$$

where $A_\sigma^2(w) = \left\langle \sigma\left(w \cdot x\right), y(x)\right\rangle^2$, $w \in \mathbb{R}^d$, $w \in \mathcal{N}\left(0, \frac{1}{d} I_d\right)$. This inner product can be calculated explicitly. For a general (normalized) probabilist Hermite polynomial, using the Hermite Rodrigues formula

together with high-order integration by parts (as a generalization of Stein's identity) produces

$$\mathbb{E}_{x\sim\mathcal{N}(0,I_d)}\left[\hat{H}e_n\left(w_*^\top x\right)\sigma\left(w^\top x\right)\right] = \frac{\left(w_*^\top w\right)^n}{n!}\mathbb{E}_{x\sim\mathcal{N}(0,I_d)}\left[\sigma^{(n)}\left(w^\top x\right)\right]$$

$$= \frac{\left(w_*^\top w\right)^n}{n!}\mathbb{E}_{t\sim\mathcal{N}(0,|w|^2)}\left[\sigma^{(n)}(t)\right]$$

assuming $|w_*| = 1$, where $\sigma^{(n)}$ stands for the the $n$-th derivative of $\sigma$. From here on, we simply use erf's third derivative, giving us straightforward Gaussian integrals to calculate that produce

$$\langle \text{erf}\left(w\cdot x\right), y\left(x\right)\rangle^2 = \frac{4}{9\pi}\left(\frac{w\cdot w_*}{\sqrt{1+2|w|^2}}\right)^6 \tag{35}$$

Next, assuming that all components of $w$ perpendicular to $w_*$ are weakly fluctuating, we can approximate

$$\frac{w\cdot w_*}{\sqrt{1+2|w|^2}} \approx \frac{w\cdot w_*}{\sqrt{1+2\left(w_*\cdot w\right)^2+2\sum_\perp\mathbb{E}_{w\cdot w_\perp}\left[\left(w\cdot w_\perp\right)^2\right]}} \approx \frac{w\cdot w_*}{\sqrt{3+2\left(w_*\cdot w\right)^2}},$$

where $\sum_\perp$ denotes a sum over an orthonormal basis in weight space, $w_\perp$, which are orthogonal to $w_*$. Using $\mathbb{E}_{w\sim\mathcal{N}(0,\frac{1}{d}I_d)}\left[g(w\cdot w_*)\right] = \mathbb{E}_{\beta\sim\mathcal{N}(0,\frac{1}{d})}\left[g(\beta)\right]$ for $\beta = w\cdot w_*$ with a unit vector $w_*$, and for any function $g$, we can apply the above approximation to (34) to get

$$t_* \approx \frac{9\alpha\pi\mathbb{E}_{\beta\sim\mathcal{N}(0,\frac{1}{d})}\left[\exp\left(\frac{2}{9\pi N}t_*^2\left(\frac{\beta}{\sqrt{3+2\beta^2}}\right)^6\right)\right]}{4\mathbb{E}_{\beta\sim\mathcal{N}(0,\frac{1}{d})}\left[\left(\frac{\beta}{\sqrt{3+2\beta^2}}\right)^6\exp\left(\frac{2}{9\pi N}t_*^2\left(\frac{\beta}{\sqrt{3+2\beta^2}}\right)^6\right)\right]}$$

If we then denote

$$S\left(\beta;t_*\right) = \beta^2 - \frac{4t_*^2}{9\pi N d}\left(\frac{\beta}{\sqrt{3+2\beta^2}}\right)^6, \tag{36}$$

we end up with

$$t_* \approx \frac{9\alpha\pi}{4}\cdot\frac{\int d\beta \exp\left(-\frac{d}{2}S\left(\beta;t_*\right)\right)}{\int d\beta\left(\frac{\beta}{\sqrt{3+2\beta^2}}\right)^6\exp\left(-\frac{d}{2}S\left(\beta;t_*\right)\right)}, \tag{37}$$

The $t_*$, which (approximately) solves (37), can then be used to compute the scaling of $P_*$.

Following the derivation for the upper bound (26), recall that we have

$$P_* \propto -2\kappa\log\left(e^{-t_*\alpha}\mathbb{E}_\Theta\left[e^{t_*A_f}\right]\right) \tag{38}$$

for $k \approx 1$. Following (31), we integrate out the readout weights and utilize our calculation, culminating in (37), to get

$$\mathbb{E}_\Theta\left[e^{t_*A_f}\right] = \mathbb{E}_W\left[e^{\frac{1}{2N}\sum_{i=1}^N t_*^2 A_{\sigma,i}^2(W)}\right]$$

$$= \prod_{i=1}^N\mathbb{E}_{w_i\sim\mathcal{N}(0,\frac{1}{d}I_d)}\left[e^{\frac{1}{2N}t_*^2 A_\sigma^2(w_i)}\right]$$

$$\approx \left[\int d\beta\exp\left(-\frac{d}{2}S\left(\beta;t_*\right)\right)\right]^N. \tag{39}$$

Finally, we arrive at the following expression for sample complexity

$$P_* \propto 2\kappa \left[ t_* \alpha - N \log \left( \int d\beta \exp \left( -\frac{d}{2} S(\beta; t_*) \right) \right) \right] \tag{40}$$

To evaluate it, we numerically solve for $t_*$ in (37). The resulting $t_*$ is then used to numerically evaluate the one-dimensional integral in (39) over $\beta = w \cdot w_*$. This produces the sample complexity scale in (40). These computational results are shown in Fig. 2, including predictions both for network output and weight distribution.

We can gain further insight by interpreting (39) as the partition function (i.e., the normalization factor) of a $t_*$-dependent, non-Gaussian distribution. This distribution can be understood as representing the required weight configuration for achieving good alignment. While good alignment is improbable under the prior, this new distribution makes it attainable. If this distribution represents the optimal path to alignment within the prior, it is reasonable to infer that it will correspond to the FL patterns in the posterior. Indeed, as illustrated in Fig. 2, this is precisely the case, as the posterior weight distribution is closely approximated by the prior alignment distribution we predict. It is this observation that motivates the reasoning in the following sections, by guessing various FL patterns that result in good alignment in the prior, we can deduce the actual distribution in the posterior. We can compare different guesses and determine which of the patterns is most likely to emerge.

## C   VARIATIONAL ESTIMATE OF $E(\alpha)$

As the explicit expression for the energy, $E(\alpha) = -\log p_{A_f}(\alpha)$, is intractable, we turn to computing a variational estimate for this quantity.

### C.1   DISTRIBUTION ANALYSIS IN PRE-ACTIVATION SPACE

We first rewrite $p_{A_f}$ as a distribution on the pre-activations, rather than the network weights. Following A.3, the upper bound for the prior probability is given by

$$\mathrm{Pr}_{p_0} [A_f \geq \alpha] \leq p_{A_f}(\alpha) := \int d\Theta \, p(\Theta) \, \delta(A_{f_\Theta} - \alpha). \tag{41}$$

Replacing the delta function with its Fourier representation, $p_{A_f}(\alpha) = \frac{1}{2\pi} \int dt \, e^{-it\alpha} \hat{p}_{A_f}(t)$, $\hat{p}_{A_f}(t) = \mathbb{E}_\Theta [\exp(itA_f)]$, we obtain

$$p_{A_f}(\alpha) = \frac{1}{2\pi} \int dt \int d\Theta \, p(\Theta) \exp \left( it \left( \sum_{i=1}^{N_{L-1}} w_i^L \left\langle y, \sigma(h_i^{L-1}) \right\rangle - \alpha \right) \right). \tag{42}$$

We can simplify this integral by integrating out the readout weights in the readout layer, as done in (31), which results in

$$p_{A_f}(\alpha) = \frac{1}{2\pi} \int dt \int d\Theta^{L-1} p(\Theta^{L-1}) \exp \left( -\frac{\sigma_L^2}{2N_{L-1}} \sum_{i=1}^{N_{L-1}} t^2 \left\langle y, \sigma(h_i^{L-1}) \right\rangle^2 - it\alpha \right) \tag{43}$$

By comparison, in (30) $t$ was real, accounting for the difference in sign. Integrating out $t$ we obtain

$$p_{A_f}(\alpha) = \tag{44}$$

$$= \frac{1}{\sqrt{2\pi}} \int d\Theta^{L-1} p\left(\Theta^{L-1}\right) \exp\left(-\frac{\alpha^2}{\frac{2\sigma_L^2}{N_{L-1}} \sum_{i=1}^{N_{L-1}} \left\langle y, \sigma\left(h_i^{L-1}\right)\right\rangle^2} + \frac{1}{2} \log \frac{1}{\frac{\sigma_L^2}{N_{L-1}} \sum_{i=1}^{N_{L-1}} \left\langle y, \sigma\left(h_i^{L-1}\right)\right\rangle^2}\right)$$

$$= \int d\Theta^{L-1} p\left(\Theta^{L-1}\right) \mathcal{N}(\alpha | 0, \kappa_A(h^{L-1}(\Theta^{L-1}))),$$

where the variance $\kappa_A$ is given by

$$\kappa_A\left(h^{L-1}\left(\Theta^{L-1}\right)\right) = \frac{\sigma_L^2}{N_{L-1}} \sum_{i=1}^{N_{L-1}} \left\langle y, \sigma\left(h_i^{L-1}\left(\Theta^{L-1}\right)\right)\right\rangle^2 \tag{45}$$

and $h_i^{L-1}$ are the final layer's pre-activations, which are themselves determined by the weights of the first $L-1$ layers of the network $\Theta^{L-1}$. We now perform a change of variables by enforcing pre-activations through a delta function and then rewriting it in terms of its Fourier representation. We then obtain

$$p_{A_f}(\alpha) = \int \mathcal{D}h \int dW \int \mathcal{D}m \, \exp\left(-\sum_{l=1}^{L-1}\sum_{i=1}^{N_l}\sum_{j=1}^{N_{l-1}} \frac{N_{l-1}}{2\sigma_l^2} \left(W_{ij}^l\right)^2\right)$$

$$\times \exp\left(i\sum_{l=1}^{L-1}\sum_{i=1}^{N_l}\left[\left\langle m_i^l, h_i^l\right\rangle - \left\langle m_i^l, \sum_{j=1}^{N_{l-1}} W_{ij}^l \sigma(h_j^{l-1})\right\rangle\right]\right) \mathcal{N}(\alpha | 0, \kappa_A\left(h^{L-1}\right)) \tag{46}$$

$$=: \int \mathcal{D}h \, \tilde{p}(h) \, \mathcal{N}\big(\alpha \mid 0, \kappa_A(h^{L-1})\big).$$

Here $\mathcal{D}h$ denotes the path integral over the pre-activation functions $h$ for all layers and neurons, and $\mathcal{D}m$ is the path integral over its Fourier conjugate variables (see (Ringel et al., 2025, Sec. 3.1) for further details). We further specify that the integration measures are normalized to include the appropriate powers of $\pi$. When $l = 1$ we replace $\sigma(h_j^{l-1})$ with $x$. We next integrate out the weights $W$ to obtain

$$\tilde{p}(h) = \int \mathcal{D}m \, \exp\left(-\sum_{l=1}^{L-1}\sum_{i=1}^{N_l}\sum_{j=1}^{N_{l-1}} \frac{\sigma_l^2}{2N_{l-1}} \left\langle m_i^l, \sigma(h_j^{l-1})\right\rangle^2 + i\sum_{l=1}^{L-1}\sum_{i=1}^{N_l} \left\langle m_i^l, h_i^l\right\rangle\right). \tag{47}$$

Finally, we integrate out the auxiliary fields $m_i^l$ to obtain

$$\tilde{p}(h) = \exp\left(-\frac{1}{2}\sum_{l=1}^{L-1}\sum_{i=1}^{N_l} \langle h_i^l, [\tilde{K}_{l-1}]^{-1}, h_i^l\rangle - \frac{1}{2} \operatorname{Tr}\log(\tilde{K}_{l-1})\right). \tag{48}$$

where the integral operator

$$\tilde{K}_{l-1}(x, x') = \frac{\sigma_l^2}{N_{l-1}} \sum_{j=1}^{N_{l-1}} \sigma(h_j^{l-1}(x))\sigma(h_j^{l-1}(x'))$$

can be understood as the fluctuating $h^{l-1}$-dependent kernel and $[\tilde{K}_{l-1}]^{-1}$ is its inverse w.r.t. the measure $d\mu_x$. The above is an exact rewriting of the probability distribution showing that, conditioned on previous layers, each hidden layer has Gaussian pre-activations. Inspired by standard statistical mechanics notation, we define the for $\tilde{p}$ the Hamiltonian $H_{\tilde{p}}(h) = \frac{1}{2}\sum_{l=1}^{L-1}\sum_{i=1}^{N_l} \langle h_i^l, [\tilde{K}_{l-1}]^{-1}, h_i^l\rangle$ and the fluctuating "partition function" $Z_{\tilde{p}}$ such that $\log Z_{\tilde{p}}(h) = \frac{1}{2}\sum_{l=1}^{L-1} \operatorname{Tr}\log(\tilde{K}_{l-1})$. Thus, we can write $\tilde{p}(h) = Z_{\tilde{p}}^{-1}(h)\exp[-H_{\tilde{p}}(h)]$.

## C.2 VARIATIONAL APPROXIMATION

Turning back to the alignment density $p_{A_f}$, we can substitute the result from the previous section so that

$$p_{A_f}(\alpha) = \int \mathcal{D}h \exp\left(-\underbrace{\left(H_{\tilde{p}}(h) + \frac{\alpha^2}{2\kappa_A(h^{L-1})}\right)}_{=:-H_{p,\alpha}(h)} - \underbrace{\left(\frac{1}{2}\log\kappa_A(h^{L-1}) + \log Z_{\tilde{p}}(h)\right)}_{=:\log Z_{A_f}(h)}\right) \quad (49)$$

We are interested in computing $E(\alpha)$, which is given by

$$E(\alpha) = -\log p_{A_f}(\alpha) = -\log \int \mathcal{D}h \exp\left(-H_{p,\alpha}(h)\right)/Z_{A_f}(h). \quad (50)$$

Since the integral over $h$ is generally intractable, we turn to a variational approximation. The goal of this estimate is to find a simpler, tractable distribution $q_\alpha(h)$ that is in some sense "close" to the true pre-activation density $\exp(-H_{p,\alpha}(h))$ per alpha. We thus define our variational distribution in a similar Hamiltonian

$$q_\alpha(h) = \frac{1}{Z_{q,\alpha}} \exp(-H_{q,\alpha}(h)), \quad (51)$$

where $H_{q,\alpha}$ satisfies $\min_h H_{q,\alpha}(h) = 0$, and $Z_{q,\alpha}$ is the partition function given by $Z_{q,\alpha} = \int \mathcal{D}h \exp\left(-H_{q,\alpha}(h)\right)$. This choice results in a unique definition of $H_{q,\alpha}$, $Z_{q,\alpha}$ for each distribution $q_\alpha$. The Feynman-Bogoliubov inequality provides a rigorous definition of closeness between $\exp(-H_{p,\alpha}(h))$ and $q_\alpha(h)$. This inequality establishes an upper bound on $E(\alpha)$ which depends on the choice of $q$. The Feynman-Bogoliubov inequality Bogolubov and Jr (2009) states that for any $q$

$$-\log\left(\int \mathcal{D}h e^{-H_{p,\alpha}(h)}/Z_{A_f}(h)\right) \leq -\log Z_{q,\alpha} + \mathbb{E}_{h\sim q_\alpha}\left[H_{p,\alpha}(h) - H_{q,\alpha}(h) + \log Z_{A_f}(h)\right] \quad (52)$$

where $\mathbb{E}_{h\sim q_\alpha}[\cdot]$ is the expectation over $h$ with respect to the distribution $q_\alpha$. Recalling that $E(\alpha) = -\log p_{A_f}(\alpha)$, this inequality can be used to simplify the bound

$$E(\alpha) \leq \mathbb{E}_{h\sim q_\alpha}\left[\log(Z_{A_f}(h)/Z_{q,\alpha})\right] + \mathbb{E}_{h\sim q_\alpha}\left[H_{p,\alpha}(h) - H_{q,\alpha}(h)\right] =: \tilde{E}_q(\alpha), \quad (53)$$

We can estimate $E(\alpha)$ by taking $q_\alpha$ which minimizes the upper bound (53). In the following sections we turn to compute the upper bound. Since we consider only $\alpha \approx 1$, we drop $\alpha$ in what follows.

When $N_{L-1}$ is large, $\kappa_A$ is weakly fluctuating with $h^{L-1}$, since it is a sum of $N_{L-1}$ random variables. Denoting $\overline{\kappa}_{A_f} = \mathbb{E}_{h\sim\tilde{p}(h)}\left[\kappa_A\left(h^{L-1}\right)\right]$, we have

$$\mathbb{E}_{h\sim q_1}\left[H_{p,1}(h) - H_{q,1}(h)\right] \approx \frac{1}{2\overline{\kappa}_A} + \mathbb{E}_{h\sim q}\left[H_{\tilde{p}}(h) - H_q(h)\right] \quad (54)$$

to leading order (where $q \equiv q_1$), so that

$$\tilde{E}_q(1) \approx \frac{1}{2\overline{\kappa}_A} + \mathbb{E}_{h\sim q}\left[H_{\tilde{p}}(h) - H_q(h)\right] + \mathbb{E}_{h\sim q_\alpha}\left[\log(Z_{A_f}(h)/Z_{q,\alpha})\right]. \quad (55)$$

We show in C.4 that the expectation of log term- $\mathbb{E}_{h\sim q_\alpha}\left[\log(Z_{A_f}(h)/Z_{q,\alpha})\right]$ is subleading and thus can be neglected. In essence, our variational ansatzes constrain deviations from the GP to a sub-extensive subset of feature directions, such that the resulting corrections do not scale extensively with system size or parameter count. Moreover, since these corrections enter logarithmically, their contribution becomes further subleading. Thus, we have

$$\tilde{E}_q \approx \frac{1}{2\overline{\kappa}_A} + \mathbb{E}_{h\sim q}\left[H_{\tilde{p}}(h) - H_q(h)\right] \quad (56)$$

The problem has now been reduced to minimizing $\tilde{E}_q$ with respect to a choice of variational distribution $q(h)$. If we find the optimal $q_*$ that minimizes $\tilde{E}_q$, we obtain our best estimate for the energy, when $\alpha \approx 1$,

$$E(\alpha \approx 1) \approx \tilde{E}_{q_*}. \quad (57)$$

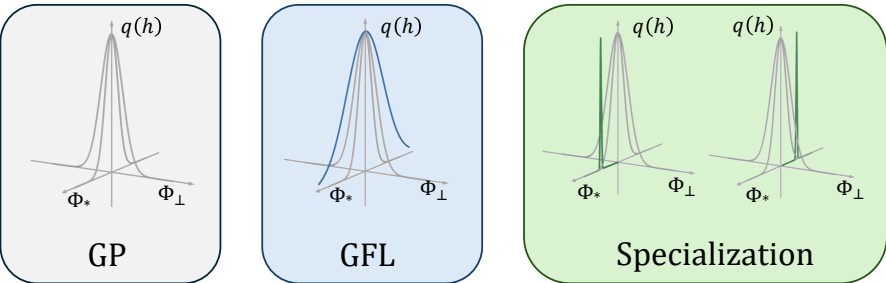

Figure 4: Schematic illustration of different candidate feature learning patterns per neuron.

Our final task is to compute the variational energy $\tilde{E}_q(\alpha)$ given our variational density $q$. Although we work with Gaussian $q$'s that are layer-wise and neuron-wise decoupled, this computation is nevertheless challenging due to the presence of $h^l$, dependence on $[\tilde{K}_l]^{-1}$, and the trace term $\mathrm{Tr}\log(\tilde{K}_l)$. In the spirit of kernel adaptation Ringel et al. (2025), we assume the fluctuations of $\tilde{K}_l$ are sufficiently small and only keep the leading order dependence which dominates $\tilde{E}_q(\alpha)$. In this context, this means replacing the fluctuating kernel $\tilde{K}_l$ with its expectation with respect to $q$ and $K_l$ given by

$$K_l\left(x, x'\right) = \frac{\sigma_{l+1}^2}{N_l} \sum_{i=1}^{N_l} \mathbb{E}_{h_i^l \sim q(h^l)}[\sigma(h_i^l\left(x\right))\sigma(h_i^l\left(x'\right))]. \tag{58}$$

Following this substitution, we obtain

$$H_{\tilde{p}}(h) = \frac{1}{2} \sum_{l=1}^{L-1} \sum_{i=1}^{N_l} \langle h_i^l, [K_{l-1}]^{-1} h_i^l \rangle \tag{59}$$

and obtain our final tractable expression for the variational energy

$$\tilde{E}_q \propto \frac{1}{2} \langle y, K_{L-1}, y \rangle^{-1} + \frac{1}{2} \mathbb{E}_{h \sim q} \left[ \sum_{l=1}^{L-1} \sum_{i=1}^{N_l} \langle h_i^l, K_{l-1}^{-1}, h_i^l \rangle - H_q\left(h\right) \right]. \tag{60}$$

Since $\tilde{p}$ is fully decoupled layer and neuron-wise, we similarly take a decoupled variational ansatz so that $q(h) = \prod_{l=1}^{L-1} \prod_{i=1}^{N_l} q_{l,i}(h_i^l)$. Further taking $q_{l,i}$ to be Gaussian with mean $\mu_{l,i}$ and variance $Q_{l,i}$, we obtain

$$\tilde{E}_q = \frac{1}{2} \langle y, K_{L-1}, y \rangle^{-1} + \frac{1}{2} \sum_{l=1}^{L-1} \sum_{i=1}^{N_l} \left\{ \mathbb{E}_{h_i^l \sim q_{l,i}} \left[ \left\langle h_i^l, [K_{l-1}]^{-1} - Q_{l,i}^{-1}, h_i^l \right\rangle \right] + \left\langle \mu_{l,i}, Q_{l,i}^{-1}, \mu_{l,i} \right\rangle \right\} \tag{61}$$

Minimizing the above over all Gaussian $q$'s results in complex non-linear equations, which we wish to avoid. Instead, we propose minimizing over a small subset of $q$'s, which we refer to as FL patterns. We look to minimize $\tilde{E}_q$ over this finite set and take that to be our variational energy. A summary of all patterns can be found in Fig. 4.

### C.3.1 Comparison to Adaptive Kernel Approximation

The similarities and differences between the approach developed here and the adaptive methods discussed in Seroussi et al. (2023a); Ringel et al. (2025) are worth examining. In those works, $\tilde{p}$ is similarly approximated

using a mean-field expansion, with two key differences: The chosen mean and the order of expansion around the mean. The fluctuating kernel in these works $\tilde{K}_{l-1}$ is expanded around its *self-consistent mean* $K_{l-1|\tilde{p}}$, rather than the variational expectation introduced here. The self-consistent mean kernel is defined as

$$K_{l>0|\tilde{p}}(x, x') = \frac{\sigma_{l+1}^2}{N_l} \sum_{i=1}^{N_l} \mathbb{E}_{h_i^l \sim \tilde{p}(h)} \left[ \sigma(h_i^l(x)) \, \sigma(h_i^l(x')) \right], \tag{62}$$

$$K_{0|\tilde{p}}(x, x') = \frac{\sigma_0^2}{d} \, x \cdot x'.$$

We define the kernel fluctuations relative to this mean by $\Delta K_{l-1}$, such that $\tilde{K}_{l-1} = K_{l-1|\tilde{p}} + \Delta K_{l-1}$. The adaptive approximations assume that these fluctuations are weak, i.e., $\Delta K_{l-1} \ll K_{l-1|\tilde{p}}$, and retain all terms up to order $\mathcal{O}(\Delta K_{l-1})$, whereas in this work we consider only the zeroth-order term in $\Delta K_{l-1}$. Taking such an approximation in the self-consistent setting would yield the GP distribution, since no feature-learning corrections would be taken into account. In our proposed methodology, because the variational expectation already incorporates feature-learning effects, the simpler expansion can instead be employed.

## C.4 Justification for Neglecting the Partition Function Ratio

Here we argue that the logarithmic ratio of partition functions, $\mathbb{E}_{h \sim q_\alpha} \left[ \log(Z_{A_f}(h)/Z_q) \right]$, is sub-leading relative to the variational energy term $\tilde{E}_q$ (defined in Sec. C.2). We show that this holds for all feature learning patterns considered in this work (Sec. 5.1): GP, GFL, and specialization. Substituting the definition of $Z_{A_f}(h)$ we obtain:

$$\mathbb{E}_{h \sim q_\alpha} \left[ \log Z_{A_f}(h) \right] = \mathbb{E}_{q_\alpha} \left[ \frac{1}{2} \log \kappa_A(h^{L-1}) \right] + \frac{1}{2} \sum_{l=1}^{L-1} \mathrm{Tr} \, \mathbb{E}_{q_\alpha} \left[ \log(\tilde{K}_{l-1}) \right] \tag{63}$$

We assume that for sufficiently wide networks the fluctuations are weak, so that at least in terms of scales we can approximate

$$\dots \approx \log \mathbb{E}_{q_\alpha} \left[ \frac{1}{2} \kappa_A(h^{L-1}) \right] + \frac{1}{2} \sum_{l=1}^{L-1} \mathrm{Tr} \log \mathbb{E}_{q_\alpha} \left[ (\tilde{K}_{l-1}) \right]$$

So that we obtain

$$\mathbb{E}_{h \sim q_\alpha} \left[ \log Z_{A_f}(h) \right] \approx \frac{1}{2} \sum_{l=1}^{L-1} \mathrm{Tr} \log K_{l-1} + \frac{1}{2} \log \overline{\kappa}_A \tag{64}$$

Recalling that $\tilde{E}_q$ scales linearly with $\overline{\kappa}_A^{-1}$, we can indeed justify neglecting the term $\log \overline{\kappa}_A$ as it is subleading for small $\overline{\kappa}_A$. We will show that the remaining terms can be neglected as well. Using the eigendecomposition of the kernel (58), $K_{l-1}(x, x') = \sum_{k_l} \lambda_{k_l} \Phi_{k_l}(x) \Phi_{k_l}(x')$, the Gaussian integral for a single neuron in layer $l$ evaluates to $\prod_{k_l} \sqrt{2\pi \lambda_{k_l}}$. The logarithm of the full partition function is then

$$\log Z_p = \sum_{l=1}^{L-1} N_l \log \left( \prod_{k_l} \sqrt{2\pi \lambda_{k_l}} \right) =: \sum_{l=1}^{L-1} N_l \log Z_p^l, \tag{65}$$

where we have defined $\log Z_p^l$ as the single-neuron log-partition function under the GP prior. As described in the main text, we consider variational distributions $q$ that fully factorize over layers and neurons. Thus, we can write

$$\log Z_q = \sum_{l=1}^{L-1} \sum_{i=1}^{N_l} \log \int Dh_i^l \, q_{l,i}(h_i^l) =: \sum_{l=1}^{L-1} \sum_{i=1}^{N_l} \log Z_q^{l,i}. \tag{66}$$

Combining these results, the log-ratio of the partition functions decomposes into a sum over per-neuron contributions. We aim to show that, for each neuron, this term is sub-dominant compared to the corresponding term in the variational energy $\tilde{E}_q$. Recall (61) that

$$\tilde{E}_q = \frac{1}{2}\langle y, K_L^{-1}, y \rangle + \sum_{l=1}^{L-1} \sum_{i=1}^{N_l} \Delta_{l,i}, \tag{67}$$

where the per-neuron cost $\Delta_{l,i}$ is

$$\Delta_{l,i} := \frac{1}{2} \mathbb{E}_{h \sim q_{l,i}} \left[ \langle h, K_{l-1}^{-1} - Q_{l,i}^{-1}, h \rangle + \langle \mu_{l,i}, Q_{l,i}^{-1}, \mu_{l,i} \rangle \right]. \tag{68}$$

Our goal is to show that, for each feature learning pattern, $\Delta_{l,i} + (\log Z_q^{l,i} - \log Z_p^l)$ scales like $\Delta_{l,i}$.

CASE-BY-CASE ANALYSIS

1. **Specialization:** The variational distribution is a delta function

$$q_{l,i}(\langle h_i^l, \Phi_*^l \rangle) = \delta[\langle h_i^l, \Phi_*^l \rangle - \mu_{l,i}], \quad q_{l,i}(\langle h_i^l, \Phi_\perp^l \rangle) = \mathcal{N}(0, \langle \Phi_\perp^l, K_{l-1}, \Phi_\perp^l \rangle). \tag{69}$$

The parition function is given by- $\log Z_q = \sum_{l=1}^{L-1} \sum_{k_l, \Phi_{k_l} \neq \Phi_*} \log \sqrt{2\pi \lambda_{k_l}}$

- **Variational Cost:** The cost term evaluates to $\Delta_{l,i} = \frac{1}{2}\mu_{l,i}^2 \langle \Phi_*, K_{l-1}^{-1}, \Phi_* \rangle$. In typical high-dimensional settings, $\langle \Phi_*, K_{l-1}^{-1}, \Phi_* \rangle$ scales at least with $d$, so $\Delta_{l,i} = \mathcal{O}(d)$.
- **Partition Function Ratio:** In all but the feature direction, the normalization factor of the variational estimate and the normalization factor of $p$ is equal, and thus cancels out. The only difference is in the feature direction, where we obtain exactly $\log Z_p^l / Z_q^l = -\frac{1}{2} \sum_{i=1}^{N_l} \log(\Delta_{li})$
- **Conclusion:** we thus obtain $Z_p^l / Z_q^l \ll \sum_{i=1}^{N_l} \Delta_{il}$.

2. **Gaussian Feature Learning (GFL):** We have $h \sim \mathcal{N}(0, Q_{l,i})$, where $Q_{l,i}$ shares the same eigenbasis as $K_{l-1}$, with $\lambda'_{k_l}$ identical to $\lambda_{k_l}$ except for the feature $\Phi_*$, where $\lambda'_* = D\lambda_*$.

- **Variational Cost:** The cost is $\Delta_{l,i} = \frac{1}{2}\text{Tr}(K_{l-1}^{-1} Q_{l,i} - I)$. Since only one eigenvalue differs, this sum reduces to $\Delta_{l,i} = \frac{1}{2}(D - 1)$.
- **Partition Function Ratio:** The log-ratio $\log(Z_q^{l,i}/Z_p^l)$ becomes the log-ratio of the determinants, which is $\frac{1}{2}\log(\det Q_{l,i} / \det K_{l-1}) = \frac{1}{2}\log(D)$.
- **Conclusion:** For large enhancement factors $D$, the linear scaling of $\Delta_{l,i} = \mathcal{O}(D)$ dominates the logarithmic scaling of the partition function term, $\mathcal{O}(\log D)$.

3. **GP / Lazy Regime:** This case is equivalent to GFL with $D = 1$.

- **Conclusion:** Substituting $D = 1$ into the GFL results, we find that both $\Delta_{l,i} = 0$ and the log-partition function term is $\log(1) = 0$. The relationship holds trivially.

## C.5 KERNEL FEATURE PROPAGATION

Since the cost of each layer depends on the kernel of the previous layer, an important element of our heuristic is understanding how the choice of pattern in the previous layer affects the kernel and its spectrum. Recall that we denote the pre-activation of the current layer by $h^l(x)$, and of the previous layer by $h^{l-1}(x)$, where $h^{l-1}(x)$ is the width $N_{l-1}$ pre-activation vector of the previous layer. Let $h_i^{l-1}$ be distributed according to

some candidate Gaussian distribution $q_i'(h^{l-1})$. We wish to understand how this choice of $q_i'(h^{l-1})$ affects the kernel of the next layer given by

$$K_{l-1}(x, x') = N_{l-1}^{-1} \sum_{i=1}^{N_{l-1}} \mathbb{E}_{h_i^{l-1}(x) \sim q_i}[\sigma(h_i^{l-1}(x))\sigma(h_i^{l-1}(x'))] =: \sum_{i=1}^{N_{l-1}} K_{l-1,i}(x, x'), \quad (70)$$

where we took $\sigma_l^2 = 1$ for brevity.

**Claim (i): Neuron specialization creates a spectral spike.** Assume we have an operator of the form $K(x, x') = A(x, x') + c\sigma(\Phi(x))\sigma(\Phi(x'))$ for some constant $c$. We wish to understand the behavior of the RKHS norm of $\sigma(\Phi)$ with respect to the above operator. Using the Sherman-Morrison formula, we obtain

$$[A + c\sigma(\Phi)\sigma(\Phi)^\top]^{-1} = A^{-1} - \frac{A^{-1}c\sigma(\Phi)\sigma(\Phi)^\top A^{-1}}{1 + c\sigma(\Phi)^\top A^{-1}\sigma(\Phi)}, \quad (71)$$

where $vv^\top$ denotes the outer product. Consequently, the RKHS norm of $\sigma(\Phi)$ with respect to $K$ is given by

$$R_K \equiv \sigma(\Phi)^\top [A + c\sigma(\Phi)\sigma(\Phi)^\top]^{-1}\sigma(\Phi) = R_A - \frac{R_A^2 c}{1 + R_A c} = \frac{R_A}{1 + R_A c} < 1/c \quad (72)$$

where $R_A$ is its RKHS norm w.r.t. $A^{-1}$. Typically, we would consider large values of $R_A$. Thus, if we have $cR_A \gg 1$, then- $\frac{R_A}{1+R_A c} \approx c^{-1}$, thereby justifying Claim (i).

**Corollary 1.** If $M$ of $N_l$ neurons are specialized in a given layer with the rest remaining Gaussian with zero mean, then we can substitute $c = N_l/M$ in the above, and obtain $R_{K_l} = N_l/M$.

**Corollary 2.** If all neurons are specialized with proportionality constant $\sqrt{\beta}$, then we can substitute $c = \beta$ in the above, and take $A = \epsilon I$ as a regularizer, with $\epsilon \to 0$, so that $R_A \to \infty$, and we obtain $R_{K_l} = \beta$.

**Claim (ii): Amplified features in the pre-activation kernel create amplified higher-order features in the post-activation kernel.** We begin by considering the Gaussian case, where $h_i^l(x) \sim \mathcal{N}(0, Q_l)$ for all $i$. Since $K_l$ is a dot product kernel for an FCN, we can expand the expectation in (58) to obtain

$$K_l(x, x') = \sum_{a=1}^{\infty} (Q_l(x, x'))^a \, F_a(x, x'), \quad (73)$$

where $Q_l(x, x')$ is a function of $x \cdot x'$ and $F_a(x, x')$ is a function of $|x|^2$ and $|x'|^2$ derived by expanding the resulting Cho and Saul kernel associated with the activation function $\sigma$ (Cho and Saul, 2009, §2.3). We further assume that $Q_l(x, x) \sim \mathcal{O}(1)$+small fluctuations. This is a reasonable assumption to make for Gaussian data in high dimensions where $|x|^2 = d \pm \sqrt{d}$. For more general datasets, one should either verify this or apply layer-normalization (as done below for the case of data with power-law covariance spectrum).

Using its Mercer decomposition $Q_l(x, x') = \sum_k \lambda_k \Phi_k(x)\Phi_k(x')$, we have

$$K_l(x, x') \propto \sum_k \lambda_k \Phi_k(x)\Phi_k(x') + \sum_{k_1,k_2} \lambda_{k_1}\lambda_{k_2} \Phi_{k_1}(x)\Phi_{k_2}(x)\Phi_{k_1}(x')\Phi_{k_2}(x') + ... \quad (74)$$

Next, we ask how enhancing a certain $\lambda_k$ by a factor $D$ affects the post-activation kernel $K_l(x, x')$. Let $\lambda_{k_*}$ be some eigenvalue of $Q_l$ corresponding to the eigenfunction $\Phi_*(x)$. It can be potentially adapted via GFL ($\lambda_{k_*} \to D\lambda_*$ where $D\lambda_* < 1$). Then the $K_{l,i}$ RKHS norm of any feature of the form $\sum_{n=1}^m a_n \Phi_{\lambda_*}^n(x)$ scales like $\mathcal{O}((\lambda_* D)^{-m})$ (assuming $a_m = \mathcal{O}(1)$). Indeed, consider the case $m = 2$ and examine

$$K_l(x, x') \propto \sum_k D_k \lambda_k \Phi_k(x)\Phi_k(x') + \sum_{k_1,k_2} D_{k_1} D_{k_2} \lambda_{k_1}\lambda_{k_2} \Phi_{k_1}(x)\Phi_{k_2}(x)\Phi_{k_1}(x')\Phi_{k_2}(x') + ... \quad (75)$$

with $D_k = D$ when $\lambda_k = \lambda_*$ and $D_k = 1$ otherwise. The argument proceeds by treating the term $\Phi_*^m(x)\Phi_*^m(x')$ as a rank-1 spike. Substituting $K_l$ for $K$, $(D\lambda_*)^m$ for $c$, and $\Phi^m$ for $\sigma(\Phi)$ in Claim (i), we obtain $R_{K_l} = (R_A^{-1} + (D\lambda_*)^m)^{-1}$.

The important observation/assumption here is that, for sufficiently large $D$, $R_A(D\lambda_*)^m \gg 1$ so that $R_{K_l} \approx (D\lambda_*)^{-m}$. This is supported numerically in Figs. 5 and 6. Our analytical reasoning is that the subspace effectively described by the kernel $K_l$ is a negligible portion of the total function space. This is either because the rank of $K_l$ is constrained by network width $N_l$ or its spectrum $\lambda_k$ exhibits rapid decay. Our analytical rationalization for this is as follows. Either because in a real network, the rank of the kernel $K_l$ is bounded by the network width $N_l$ or because the spectrum $\lambda_k$ falls down quickly, the effective span of the first term and subsequent terms covers only a negligible fraction of the total function space. In this large function space, it is unlikely that a feature like $\Phi_*^m$ is fully contained within this span. If this is the case, its RKHS norm with respect to the base kernel would be large, motivating our observation/assumption.

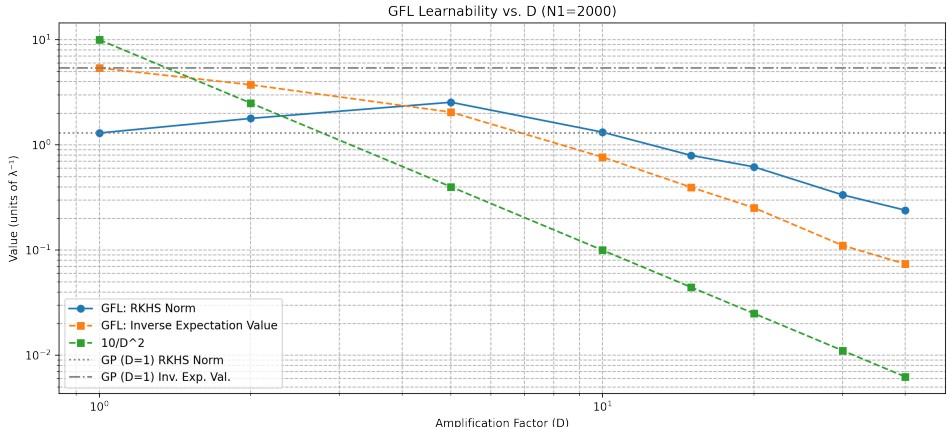

Figure 5: Demonstration GFL feature propagation. We increased the variance of the $d/2$ highest kernel mode $\Phi_*(x)$ of the first hidden layer and then measured $\langle \Phi_* | K_{l=3} | \Phi_* \rangle^{-1}$ and $\langle \Phi_* | [K_{l=3}]^{-1} | \Phi_* \rangle$, where $K_{l=3}$ is the kernel of the subsequent layer. We used ReLU activations, $d = 120$, $N_1 = 2000$, $N_2 = 1000$ and random Gaussian data. This demonstrates the expected $D^{-2}$ decay and the matching between inverse expectation values and the RKHS.

In Fig. 5, we consider two ReLU-activated layers of an FCN ($L = 4$) and Gaussian i.i.d. data $x \in \mathbb{R}^d$, $x \sim \mathcal{N}(0, I_d)$. We obtain the empirical NNGP kernel after the first ReLU layer of width $N_1 = 2000$. Diagonalizing the kernel ($K_{l=2}$) on $P = 6000$ sample points $\{x_\nu\}_{\nu=1}^P$, we choose $\Phi_*(x)$ to be the $d/2$ highest eigenvalue. We then look for the best weight vector $a \in \mathbb{R}^{N_1}$ satisfying $\Phi_*(x_\nu) \approx \sum_{i=1}^{N_1} a_i \mathrm{ReLU}(W_i \cdot x_\nu)$, where $W_i$ are the input layer weights used to generate $K_{l=2}$. We draw the rows of $W \in \mathbb{R}^{N_2 \times N_1}$ i.i.d. with $W_i \sim \mathcal{N}\left(0, I/N_1 + (D-1)\hat{a}\hat{a}^\top/N_1\right)$, where $\hat{a} = a/\sqrt{a^\top a}$. This mimics the GFL effect. Finally, we compute the kernel empirical kernel following another ReLU layer and compute the RKHS norm of $\Phi_*^2(x)$ with mean removed, omitting eigenvalues which are zero up to machine precision (for $P > N_2$, $P - N_2$ such eigenvalues are to be expected).

In Fig. 6, we perform a similar analysis with the following qualitative changes. The data is chosen to be Gaussian, but non-i.i.d. with a covariance matrix whose $k$'th eigenvalue is $k^{-1.1}$. To prevent strong

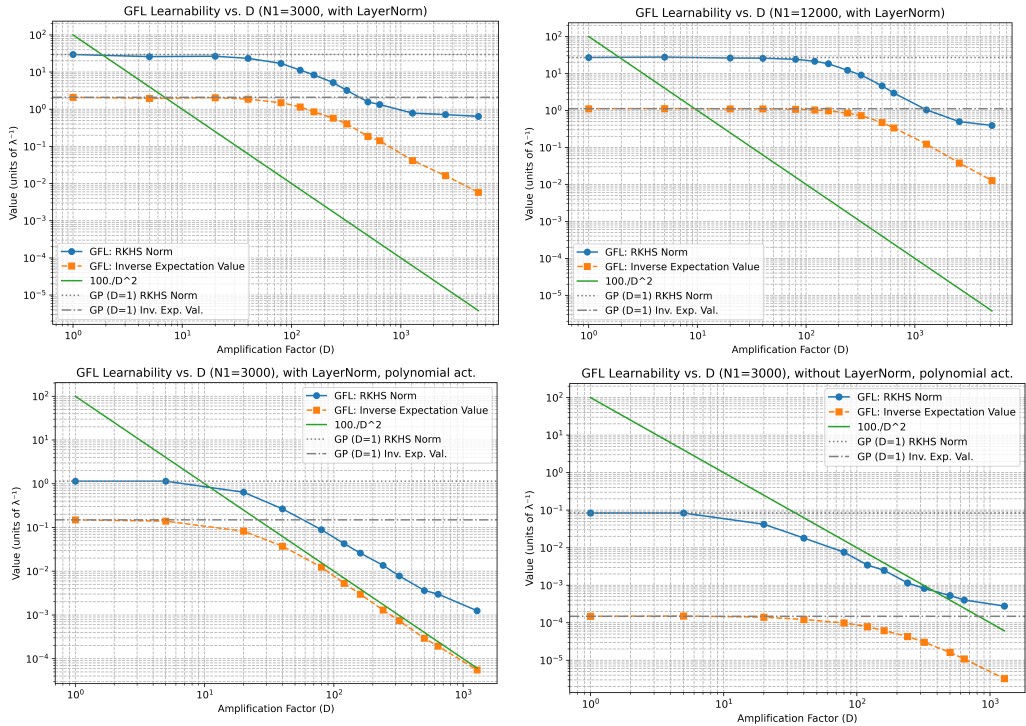

Figure 6: Similar to Fig. 5, only for Gaussian data with power-law covariance matrix with eigenvalues decaying as $k^{-1.1}$. See C.5 for further details.

fluctuations of $K_2(x,x)$ (i.e., $Q_2(x,x)$ of Claim (ii)), which are now not suppressed at $d \to \infty$ we apply layer normalization between the two activated ReLU layers. The top left panel has $d = 250, N_1 = N_2 = 3000$ and uses 4000 datapoints to sample the kernel operator and compute the spectrum used in the RKHS norm. The top right panel is a $\times 4$ scaled-up version of the left panel in terms of $N_1, N_2, P$ and $d$ and shows a longer persistence of $1/D^2$ expected scaling by, roughly, a factor of 2. The bottom panels show the behavior of polynomial activation of the type $\sigma(x) = 1 + x + x^2$, both with (left) and without (right) layer normalization.

## D  $\tilde{E}$ AND $\Delta_\ell$ IN WEIGHT SPACE

In the main text, we focused on a function space formulation for $\tilde{E}$, using pre-activations, GP distributions ($q$) for $h^l$, and kernel operators. This formulation facilitated the combination of GP-like learning mechanisms and circuit/specialization-based mechanisms. In some cases however, it is more natural to work in weight space (see also Guth et al. (2024) for a weight-space picture analogous to kernel adaptation). One such example is CNNs, for which the distribution of the input layer convolutional-patch-dependent pre-activations, is more readily described in terms of shared input weights. Another example would be an $i_0$-th neuron specializing on some $j_0$ neuron of the upstream layer, which implies a delta-function distribution around $W_{i_0 j} \sim \delta_{j,j0}$. Accordingly, we wish to express $\Delta_{l,i}$ of (68) in weight space. For simplicity, we focus on distributions with zero mean.

To this end, we consider some pre-activation in layer $l$ and write it in terms of layer weights namely, $h_{i_0}^l(x) = \sum_{i=1}^{N_{l-1}} W_{i_0 i}^l \sigma(h_i^{l-1}(x))$. For compactness, we henceforth take $W_{i_0 i}^l \to a_i, h_{i_0}^l \to h^l$. We next treat the $h_i^{l-1}$'s as independent draws from $q_{l-1,i} = q_{l-1}$, consistent with the kernel adaptation approximation, which removes correlations between pre-activation. To avoid operator algebra and work solely with matrices, we consider extremely many $(P')$ draws $\{x_\nu\}_{\nu=1}^{P'}$ from $d\mu_x$ and define the $(l-1)$-th feature matrix with entries

$$F_{i\nu}^{l-1} := \sigma(h_i^{l-1}(x_\nu)), \quad i = 1, \ldots, N_{l-1}, \quad \nu = 1, \ldots, P'$$

Note that for large $N_{l-1}$, $\left[ \left( F^{l-1} \right)^\top F^{l-1} \right]_{\mu\nu} / N_{l-1}$, via the law of large numbers, concentrates to its averages under $q$ and thus approaches $K_{l-1}(x_\mu, x_\nu)$.

We next revisit the inter-layer action for $l, l-1$ appearing Eq. (48 given by $\langle h, \tilde{K}^{-1}, h \rangle$ and note that (i) $\tilde{K}$ upon freezing $h^{l-1}$ at some typical value as done above, $\tilde{K} = F^\top F / N_{l-1}$ (ii) the inverse operation is to be understood as a pseudo inverse $((..)^+)$ which removes the contribution of the null space of this (generically) rank-$N_{l-1}$ operator. Following this and the linear relation between $h^l$ and $a$, we have that

$$\langle h, \tilde{K}_{l-1}^+, h \rangle = a^T F [F^\top F / N_{l-1}]^+ F^\top a = N_{l-1} a^\top a. \tag{76}$$

We thus obtain a reasonable result: The RKHS term, which regulates the pre-activations, is equal to the weight-decay term associated with the $a$'s that generated that $h$.

Following this rewriting, one can now repeat our variational approximation in weight space. This results in an analogous formula to Eq. 11 with $q_l$ denoting a Gaussian distribution on $a$'s ($a \sim \mathcal{N}[\bar{a}^l, \Sigma_l]$), and $K_{l-1}$ (the covariance of the $h$'s in the action) replaced by $I/N_{l-1}$ (the covariance of $a$'s in the action).

All in all we obtain the following weight space version of the contribution of the $i$'th neuron to $\tilde{E}_q$, namely

$$\Delta_{l,i} = \mathbb{E}_{a \sim \mathcal{N}(\bar{a}, \Sigma_l)} \left[ a^\top [N_{l-1} I - \Sigma_l^{-1}] a \right] + \bar{a}^T \Sigma_l^{-1} \bar{a} \tag{77}$$

# E  APPLICATION OF HEURISTICS FOR ADDITIONAL ARCHITECTURES

## E.1  REGRESSION TASKS IN TWO-LAYER NETWORKS

In the two-layer setting, an exact solution can be obtained, so we begin by comparing our heuristic approach to the exact solution. In this case, we consider both two-layer FCNs as well as CNNs with non-overlapping convolution windows. Together, these are given by

$$f(x) = \sum_{i=1}^{N_w} \sum_{j=1}^{N} w_{ij}^{(2)} \sigma(w_j^{(1)} \cdot x_i), \tag{78}$$

where $x \in \mathbb{R}^d$ is drawn from $\mathcal{N}(0, I_d)$. We take $d = N_w S$ so that $w_j, x_i \in \mathbb{R}^S$. The vector $x_i$ is given by the $((i-1)S + 1)$-th to $iS$-th coordinates of $x$. We train these networks on a polynomial target of degree $m$ given by $y(x) = \sum_{i=1}^{N_w} He_m(w_* \cdot x_i)$ where $He_m$ is the $m$-th probabilist Hermite polynomial, which is the standard polynomial choice under our choice of data measure, and $w_* \in \mathbb{R}^S$ is some normalized vector. The networks are trained via Lengevin dynamics Welling and Teh (2011b), with ridge parameter $\kappa$, quadratic weight decay, and standard scaling. For an extension to mean-field scaling, see App. E.1.1.

## E.1.1  FULLY CONNECTED NETWORKS

**Standard scaling.**  We turn to compute $\tilde{E}_q$ for a range of feature learning patterns. For this shallow FCN (and $N_w = 1$), our choice of feature learning patterns amounts to considering distributions in a single layer.

We consider the following three scenarios (though more combinations are possible): (1) all neurons are GP distributed, (2) all are GFL distributed with amplification $D$, and (3) $M$ neurons specialize on a the same feature, while those remaining are GP distributed. As the kernel of the first layer can only express linear features, the only relevant feature to be considered for the GFL and M-specialization patterns is $\Phi_*(x) = w_* \cdot x$. We compute the scale of the optimal variational energy for each pattern:

(1) **GP**: Here, $\Delta_{1,i} = 0$ since $K_{l-1} = Q_l$. In this baseline setting, learning $m > 1$ is hard since $\langle He_m | K_2 | He_m \rangle = \mathcal{O}(d^{-m})$ (see Sec. 5.2). Thus, in total, we have $\tilde{E}_{q \sim \text{GP}} \propto d^m$.

(2) **GFL**: Following (11), this pattern incurs a cost of $\Delta_{1,i} = D$ per neuron $i$, resulting in a total cost of $ND$. The $a_y$ term can be calculated utilizing Claim (ii). This leads to a $D^m$-factor decrease in the RKHS norm relative to the GP, so that $a_y \propto (d/D)^m$. In total, we find that $\tilde{E}_{q \sim \text{GFL}} \propto ND + (d/D)^m$. Minimizing w.r.t. $D$, we obtain $D_{\min} = (d^m/N)^{1/(m+1)}$, and, substituting back, we obtain $\tilde{E}_{q \sim \text{GFL}} \propto (Nd)^{\frac{m}{m+1}}$.

(3) **M-Specialization**: Following 11, this pattern incurs a cost of $\Delta_1 = M \langle \Phi_*, K_0^{-1} \Phi_* \rangle = Md$, where we denote $\Delta_l = \sum_i \Delta_{l,i}$. Utilizing Claim (i), this results in adding a spike with an $M/N$ coefficient along $\sigma(w_* \cdot x)$ in $K_1$ appearing in $a_y$. Before this spike, $He_m(x)$ only had overlaps with the $m$-th order Taylor expansion of the kernel, leading to a $d^{-m}$ scaling. However, since $\sigma(w_* \cdot x)$ has an $\mathcal{O}(1)$ overlap with $He_m(x)$, so that $a_y \propto N/M$, we obtain $\tilde{E}_{q \sim \text{M-Sp}} \propto dM + N/M$. Minimizing further over $M$, the number of specializing neurons leads to $M_{\min} = \sqrt{N/d}$ and therefore $\tilde{E}_{q \sim \text{M-Sp}} \propto \sqrt{dN}$.

Now, we can compare the different feature learning patterns. Taking the most common linear scaling where $N \propto d$, the specialization scenario has the lowest variational energy. Our scaling theory then predicts an $\mathcal{O}(d)$ sample complexity as well as multimodal distribution of $w$ along $w_*$ with $\mathcal{O}(1)$ specializing neurons. Taking $m > 1$ and $N \gg d^5$, lazy learning wins and leads to $\mathcal{O}(d^m)$ complexity. When $m = 1$, GFL and M-specialization are on par for $N \propto d$. These calculations coincide with both experimental and direct LDT results, as demonstrated in Fig. 2 for networks trained on $He_3$. In terms of sample complexity, both predictions agree with experiment, with a scaling of $P_* \propto d$, as seen in Fig. 2 (b). Our heuristic approach correctly predicts the scaling of the number of specializing neurons with $N$, as seen in Fig. 2 (c). Finally, as shown in panel (a), the analytical LDT method recovers the correct pre-activation distribution, which corresponds to $q(h)$ for $q \sim$ M-Sp.

**Extension to Mean-field Scaling** Here we extend the results presented in the main text to the case of mean-field scaling. We note that the results presented for standard scaling can change when introducing mean-field parametrization, amounting here to enhancing the alignment factor by a $\chi = \mathcal{O}(N)$ factor. The GP pattern then results in a much worse energy $\tilde{E}_{GP} = Nd^3$. Revisiting the GFL pattern, we now find $\tilde{E}_{GFL} = ND + N(d/D)^m$ leading to $D \propto d^{1/(m+1)}$ and $\tilde{E}_{GFL,optimal} \propto Nd^{m/(m+1)}$. For specialization, we obtain $\tilde{E}_{sp} = dM + N^2/M$, leading to $M \propto N/\sqrt{d}$ and $\tilde{E}_{sp,optimal} \propto \sqrt{d}N$. Taking $N \propto d$, the specialization scenario again wins over for $m > 1$, with a sub-extensive ($\mathcal{O}(\sqrt{d})$) number of specializing neurons. Unlike with standard-parametrization, we see that even when taking $N \to \infty$, we remain in the rich regime. However, at least in this Bayesian finite-ridge setting, sample complexity is better with standard parametrization.

### E.1.2 CNNS WITH NON-OVERLAPPING PATCHES.

Our approach can be extended to the CNN in (**??**) with $N_w > 1$. In this case, it is better to focus on the covariance of $w_i$, namely, $\Sigma = N^{-1} \sum_{i=1}^{N} w_i w_i^T$, than on the covariance of pre-activations on each path. One can then show that the cost becomes $\Delta_1 = \mathrm{E}_{q_w} w^\top \left[ \Sigma^{-1} - I_S/S \right] w$.

(1) **GP**: In this scenario, the output kernel is given by $K_{2,\text{CNN}}(x, x') = N_w^{-1} \sum_{i=1}^{N_w} K_{2,\text{FCN}}(x_i, x'_i)$, where $K_{2,\text{FCN}}$ is the FCN kernel ($N_w = 1$). Focusing on a linear target for simplicity, we can work out the scaling of the relevant (linear) kernel feature by Taylor expanding $K_{2,\text{FCN}} = a_1 x_i \cdot x'_i / S$ to get $K_{2,\text{CNN}} = \frac{a_1}{N_w S} x \cdot x'$, with $a_1$ being some $\mathcal{O}(1)$ constant. Lazy learning then yields $\tilde{E}_{q \sim \text{GP}} = N_w S = d$, as in a FCN with no weight sharing.

(2) **GFL**: Here, we take $\Sigma = I_S / S + D w_* w_*^\top$, resulting in $\Delta_{l=1} = ND$. The leading term of the Taylor expansion now equals $K_{2,\text{CNN}} = \frac{1}{N_w S} \sum_{i=1}^{N_w} x_i^\top \Sigma x'_i$ leading to a $D/(N_w S)$ scaling of the target. All in all, we find that $\tilde{E}_{q \sim \text{GFL}} = ND + (N_w S)/D$, leading to $\tilde{E}_{q_*} = \sqrt{N M_w S}$ for the optimal $q_* \sim \text{GFL}$.

(3) **M-Specialization**: In this case, we obtain a $\Delta_{l=1} = MS$ cost. The contribution of the $M$ specializing neurons to the kernel goes as $\frac{a_1 M}{N_w N} \sum_{i=1}(w_* \cdot x)(w_* \cdot x')$, leading to $a_y = \langle y, K_L, y \rangle = \mathcal{O}(M/N_w N)$. Thus, $\tilde{E}_{q \sim \text{M-Sp}} = MS + N_w N / M$ resulting in the optimal variational energy $\sqrt{S N_w N}$.

Both GFL and M-Specialization patterns, in the proportionate limit $N \propto N_w \propto S \to \infty$, lead to $P_* \propto S^{3/2} = d^{3/4}$. This recovers results reported in Ringel et al. (2025) computed via a mean-field approach.

## E.2 CLASSIFICATION TASKS

We consider a fully connected network as defined in 78 with $N_w = 1$ and ReLU activations, trained on a $k$-parity classification task. Explicitly, this task is defined as-

$$y_k(x) = \text{sign} \prod_{j=1}^{k} x_j, \tag{79}$$

where $\eta \sim \mathcal{N}(0, 1)$, and for $i = 1, .., d$, $x_i = s_i + \mathcal{N}(0, \epsilon)$, for and $s_i$ that are i.i.d. random binary valuables in $\{\pm 1\}$. We take cross-entropy loss, and consider $k = 2$.

Here we observe that the target is given by-

$$y_2(x) = \text{sign}((x_1 + \eta_1)(x_2 + \eta_2)) = \text{sign}(\frac{1}{2}(x_1 + x_2)^2 - x_1^2 - x_2^2) \tag{80}$$

This setting is similar to the previously discussed polynomial target in the erf networks. The difference in this case is that there are multiple relevant directions, namely: $w_{*,1} = \hat{e}_1, w_{*,2} = \hat{e}_2, w_{*,1+2} = \hat{e}_1 + \hat{e}_2$, and up to multiplicative constants on the target and constant additions. As in the previous cases, we compute the scale of the optimal variational energy for each pattern, and consider the same patterns as in the two layer network. In this case, we can further consider another setting of feature learning. Rather than taking $M$ specializing neurons, and setting each of them to specialize with amplitude $\mu = 1$, we can set $\approx 1$ neuron to specialize with magnitude $\mu$. As the erf was a bounded activation, this choice was not beneficial in that case we did not need to consider it as well. Defining the last pattern as $\mu$-Specialization, and comparing to the previously discussed patterns, we obtain:

(1) **GP**: Here, $\Delta_{1,i} = 0$ since $K_{l-1} = Q_l$. As in the case of the erf network, we have $\tilde{E}_{q \sim \text{GP}} \propto d^2$.

(2) **GFL**: Here we have $\tilde{E}_{q \sim \text{GFL}} \propto ND + (d/D)^2$. Minimizing w.r.t. $D$, we obtain $D_{\min} = d^{2/3}/N^{1/3}$, and, substituting back, we obtain $\tilde{E}_{q \sim \text{GFL}} \propto (Nd)^{\frac{2}{3}}$.

(3) **M-Specialization**: $\tilde{E}_{q \sim \text{M-Sp}} \propto dM + N/M$. Minimizing further over $M$, the number of specializing neurons leads to $M_{\min} = \sqrt{N/d}$ and therefore $\tilde{E}_{q \sim \text{M-Sp}} \propto \sqrt{dN}$.

(4) **$\mu$-Specialization**: Following 11, and the second feature propagation rule, we obtain this pattern incurs a cost of $\Delta_1 = \mu^2 d$. Utilizing the first feature propagation rule, this pattern results in adding a spike

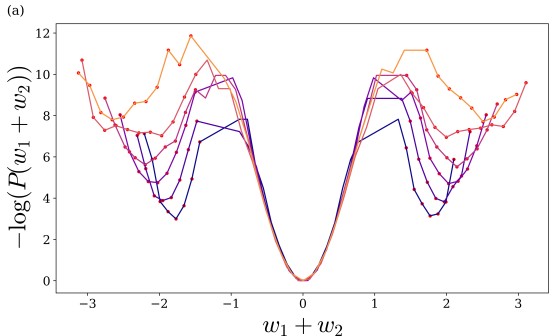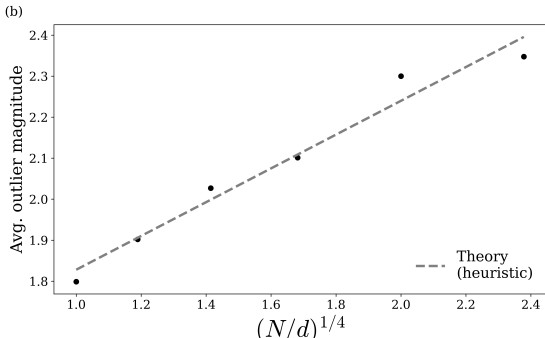

Figure 7: Emergence of specialization in two layer ReLU networks trained on classification task. In (a), the negative log distribution of the weights in the hidden layer in one of the feature directions is shown, with neurons classified as specializing appearing in red. Specifically, in this figure the distribution is shown for $w_{*,1+2}$. Although in this setting there are multiple relevant feature direction, we observe a similar $\mu$-Specialization behavior in all of these. A single direction is considered here for simplicity. In (b) the magnitude of the specializing neuron as a function of the network width is shown. Indeed, our Heuristics manage to correctly predict the scaling of the learned feature with increasing here.

of the kernel so that $a_y \propto N/\mu^2$, we obtain $\tilde{E}_{q\sim\text{M-Sp}} \propto d\mu^2 + N/\mu^2$. Minimizing further over $\mu$, the magnitude of the specializing neurons leads to $\mu_{\min} = (N/d)^{1/4}$ and therefore $\tilde{E}_{q\sim\mu\text{-Sp}} \propto \sqrt{dN}$.

There is a redundancy in the preferential patterns, as both specialization distributions result in a variational energy that scales as $P_* \propto \sqrt{Nd}$. Experimentally, we find that indeed the feature learning patterns that emerge for this case correspond to $\mu$-Specialization, with the correct scaling of the specialization magnitude with the layer width, as can be seen in 7.

### E.3 SOFTMAX ATTENTION LAYER

Here, we consider an attention block of the form

$$f(X) = \frac{1}{\sqrt{L}} \sum_{h=1}^{H} \sum_{a,b=1}^{L} @_{ab;h}(X)(w_h \cdot x^b) \tag{81}$$

$$@_{ab;h}(X) = \frac{e^{[x^a]^\top A_h x^b}}{\sum_{c=1}^{L} e^{[x^a]^\top A_h x^c}}$$

where $X \in \mathbb{R}^{L\times d}$, $A_h \in \mathbb{R}^{d\times d}$, $x^a \in \mathbb{R}^d$ is the $a$-th row of $X$, and $w_h \in \mathbb{R}^d$. Our prior over network weights is $\prod_{h=1}^{H} \mathcal{N}[0, I_{d^2}/d^2; A_h]\mathcal{N}[0, I_d/(dH); w_h]$. The only context length ($L$) dependence comes from the pre-factor of $1/\sqrt{L}$ which ensures that when $X_i^a \sim \mathcal{N}(0,1)$ then we have $f(X) = \mathcal{O}(1)$. We consider the target function $y(X) = \sum_{a,b} \frac{1}{\sqrt{L(L-1)}} x_1^a x_2^b x_3^b$.

As $H \to \infty$, the above $f(X)$ tends to a GP with $w_h$ taking the role of read-out layer weights and a kernel given by

$$K(X, X') = (dL)^{-1} \sum_{a,a',b,b'=1}^{L} (x^b \cdot (x')^{b'}) \mathbb{E}_A \left[ @_{ab}(X)@_{a'b'}(X') \right], \tag{82}$$

where we dropped the multi-head index $h$ by our i.i.d. assumption.

Our goal is to find the sample complexity and scaling of feature learning effects with context length ($L$) at finite $d$. To this end, we shall consider the following weight-space version of $\tilde{E}$ given by,

$$\tilde{E}_q = \sum_{h=1}^{H} d^2 \mathbb{E}_{A_h \sim q_h} \left[ Tr[A_h A_h^\top] - 1 \right] + \frac{1}{\langle y, K, y \rangle}. \tag{83}$$

where $K$ is computed using the $q_h$ distribution for $A$. There are several ways of obtaining this variational energy; a direct microscopic route is shown in Sec. E.3.1. Alternatively, one can use the arguments of Sec. D, previously employed in Sec. E.1.2, and trade the kernel version of $\Delta_{l,i}$ with the excess weight decay of that layer. The latter being the first term in the above formula.

We will focus on two learning patterns here: lazy learning (GP) and weight specialization. For lazy learning, we take $q_h = \mathcal{N}[0, I_{d^2 \times d^2}/d^2; A_h]$ leading, as expected, to zero contribution from the first term in $\tilde{E}_q$.

To assess the scaling of $\langle y(X), K(X, X'), y(X') \rangle$ with $L$, we make the following observations. For large $L$ and random $A_h$'s, the denominator of $@_{ab}(X)$ is $O(L)$ plus smaller fluctuations namely

$$@_{ab;h}(X) = \frac{e^{[x^a]^\top A_h x^b}}{\sum_{c=1}^{L} e^{[x^a]^\top A_h x^c}} = \frac{e^{[x^a]^\top A_h x^b}}{Z + [\sum_{c=1}^{L} e^{[x^a]^\top A_h x^c} - Z]} \tag{84}$$

$$Z = \mathbb{E}_A \left[ \sum_{c=1}^{L} e^{[x^a]^\top A_h x^c} \right] = O(L)$$

which allows an expansion in the small parameter $[\sum_{c=1}^{L} e^{[x^a]^\top A_h x^c} - Z]/Z = O(1/\sqrt{L})$. Next, we observe that functions involving $l$-sequence index functions, such as our $l = 2$ target, appear at $l - 2$ order of this expansion. Thus, we suffice with a leading order expansion

$$\langle f(X), y(X) \rangle = \frac{1}{\sqrt{L}} \sum_{h=1}^{H} \left\langle Z^{-1} \sum_{ab=1}^{L} e^{[x^a]^\top A_h x^b} w_h \cdot x^b + O(L^{-1}), \sum_{c,d=1}^{L} \frac{1}{\sqrt{L(L-1)}} x_1^c x_2^d x_3^d \right\rangle \tag{85}$$

$$= \frac{1}{\sqrt{L}} \sum_{h=1}^{H} \left\langle Z^{-1} \sum_{ab=1}^{L} e^{[x^a]^\top A_h x^b} w_h \cdot x^b + O(L^{-1}), \frac{1}{\sqrt{L(L-1)}} x_1^a x_2^b x_3^b \right\rangle = \ldots$$

where we used the fact that, under our Gaussian measure, $x_2^d x_3^d$ projects out any term which does not contain an odd power of $x_2^b$ and an odd power of $x_3^b$ and a similar consideration with $x_1^a$. Next, a direct computation gives

$$\ldots = \frac{1}{\sqrt{L^2(L-1)Z}} \sum_{h=1}^{H} Z^{-1} \sum_{ab=1}^{L} \left( [A_h]_{12}[w_h]_3 + [A_h]_{13}[w_h]_2 (1 + O(Tr[A_h A_h^\top])) + O([A_h A_h^\top A_h]_{12}[w_h]_3) + O(A^5) \right) \tag{86}$$

$$+ O(L^{-1}) = \ldots$$

Noting that under our distribution for $A$, $Tr[AA^\top] = O(1)$ and $O([A_h A_h^T A_h]_{ij}) = O([A_h]_{ij})$ we finally obtain that the typical scale of the r.h.s. the same as its leading order expansion in $A$ namely

$$\ldots \propto \frac{\sqrt{L}}{Z} \sum_{h=1}^{H} \left( [A_h]_{12}[w_h]_3 + [A_h]_{13}[w_h]_2 \right) \tag{87}$$

Finally, computing $\langle y(X), K(X, X'), y(X') \rangle$ amount to computing the 2nd moment of the above term under the prior for $A_h, w_h$ which, given that $Z = O(L)$, yields a scaling

$$\langle y, K, y \rangle \propto \frac{1}{Ld^3} \tag{88}$$

Implying that $\tilde{E}_{lazy} \propto Ld^3$.

We next consider a specializing scenario ($q_{sp.}$) where for $O(1)$ heads, $[A_h]_{12}$ has an average of the order $M/d$. Other components of $A_h$ as well as $A_h$'s of the remaining heads, remain lazy. The motivation for this choice, also observed empirically, is that together with a spike in $[w_h]_3$, one can create strong alignment in Eq. (87). Other combinations of $1, 2, 3$ indices are also possible and lead to the same scaling.

Using the above $q_{sp.}$, the excess weight-decay (first term in $\tilde{E}_q$) gets contributions only from the $O(1)$ specializing heads scaling as $d^2(M/d)^2$. One further notes that using this $A_{12}$, in Eq. 87 and subsequently in 88, yields $\langle y, K, y \rangle \propto \frac{M^2/d^2}{LdH}$ leading to $\tilde{E}_{sp.} = M^2 + HLd^3/M^2$. Optimizing the latter over $M^2$ leads to $\tilde{E}_{sp,opt} = \sqrt{LH}d^{1.5}$. This scenario is thus favorable to lazy learning up to $H = O(\sqrt{Ld^3})$.

To validate these results, we consider the above model, with $d = 8$ and $d = 16$, $H = 2$, $\kappa = 0.1$, and $P = 350, 500, 707, 1000$ data points, and a maximal context length of $180, 360, 720, 720$ respectively. Fig. 3(b) shows target alignment of the equilibrated network as a function of $\sqrt{Ld^3}/P$, demonstrating that $P = \sqrt{Ld^3}$ is the scale at which $O(1)$ alignment appears. A similar plot for the Test MSE results is shown in the right panel.

### E.3.1 DIRECT DERIVATION OF $\tilde{E}_q$ FOR A SOFTMAX BLOCK

Gathering all $A_h, w$ parameters into $\Theta$, the probability of aligment $\alpha$ is given by

$$p_{A_f}(\alpha) = \frac{1}{2\pi} \int dt \int d\Theta p(\Theta) \exp\left(it\left(\langle y, f_\Theta \rangle - \alpha\right)\right). \tag{89}$$

We can integrate out $w_h$ as follows:

$$p_{A_f}(\alpha) = \frac{1}{2\pi} \int dt \int d\Theta p(\Theta) \exp\left(it\left(\left\langle y, \frac{1}{\sqrt{L}}\sum_{h=1}^{H}\sum_{a,b=1}^{L} @_{ab;h}(x)(w_h \cdot x^b)\right\rangle - \alpha\right)\right).$$

Let $v_h(x) := \sum_{b=1}^{L}\left(\sum_{a=1}^{L} @_{ab;h}(x)\right)x^b$. Then, we can re-write

$$p_{A_f}(\alpha) = \frac{1}{2\pi} \int dt \int d\Theta p(\Theta) \exp\left(it\left(\frac{1}{\sqrt{L}}\sum_{h=1}^{H}\langle y, w_h \cdot v_h\rangle - \alpha\right)\right)$$

$$= \frac{1}{2\pi} \int dt \int d\Theta p(\Theta) \exp\left(it\left(\frac{1}{\sqrt{L}}\sum_{h=1}^{H} w_h \cdot \langle y, v_h\rangle - \alpha\right)\right)$$

where we're using an unusual notation, regarding $\langle y, v_h \rangle$ as a vector of $L_2$ inner products. Using the prior we continue:

$$= \frac{1}{2\pi} \int dt \exp(-it\alpha) \prod_{h=1}^{H} \int dA_h p(A_h) \int dw_h p(w_h) \exp\left(\frac{it}{\sqrt{L}} w_h \cdot \langle y, v_h \rangle\right)$$

$$= \frac{1}{2\pi} \int dt \exp(-it\alpha) \prod_{h=1}^{H} \int dA_h p(A_h) \exp\left(-\frac{1}{2} \frac{t^2}{LdH} \langle y, v_h \rangle^\top \langle y, v_h \rangle\right)$$

$$= \frac{1}{2\pi} \int dA\, p(A) \int dt \exp\left(-\frac{1}{2} \frac{t^2}{LdH} \sum_{h=1}^{H} \langle y, v_h \rangle^\top \langle y, v_h \rangle - it\alpha\right)$$

When we integrate out $t$ we get

$$p_{A_f}(\alpha) = \tag{90}$$

$$= \frac{1}{\sqrt{2\pi}} \int dA\, p(A) \exp\left(-\frac{\alpha^2}{\frac{2}{LdH} \sum_{h=1}^{H} \langle y, v_h \rangle^\top \langle y, v_h \rangle} + \frac{1}{2} \log \frac{1}{\frac{1}{LdH} \sum_{h=1}^{H} \langle y, v_h \rangle^\top \langle y, v_h \rangle}\right)$$

Let's denote

$$\kappa(A) := \frac{1}{LdH} \sum_{h=1}^{H} \langle y, v_h \rangle^\top \langle y, v_h \rangle = \frac{1}{LdH} \sum_{h=1}^{H} \sum_{i=1}^{d} \langle y, [v_h]_i \rangle^2 .$$

Then we have

$$p_{A_f}(\alpha) = \int dA\, p(A) \frac{1}{\sqrt{2\pi\kappa(A)}} \exp\left(-\frac{\alpha^2}{2\kappa(A)}\right)$$

$$= \int dA\, p(A) \mathcal{N}(0, \kappa(A); \alpha)$$

Next, we take a layer-decoupling approximation, similar in spirit to kernel adaptation, where we assume $\kappa(A)$ does not fluctuate out of scale, with $A$. We thus trade $\kappa(A)$ by its average, which yields our $a_y$ term, $\alpha^2/\langle y, K, y \rangle$. As in the original derivation, we expect $a_y$ to be extensive ($O(d, N, ..)$) and hence its log, or equivalently the above $1/\sqrt{2\pi\kappa(A)}$ prefactor of the exponent is negligible. The action ($S$), or negative log-probability associated with $p(A)e^{-\alpha^2/(2\kappa(A))}$ is now given by $S = (d^2 Tr[A^T A] + \alpha^2/\kappa(A))/2$. Using a variational principle on this action with a Gaussian $q$, one obtains the above variational energy, with the $-d^2$ contribution coming from the entropy term $\int dA q(A) \log(q(A))$ in the KL-divergence used in the variational approach (similarly to how $-H_{q,\alpha}(h)$ emerged in sec. C.2)

# F   EXPERIMENTAL DETAILS

## F.0.1   TWO-LAYER FCN- EXPERIMENTAL DETAILS

In panels (a) and (b) of Fig. 2, the experimental data points were computed by training an ensemble of 300 networks. For panel (a), all experiments used $P = 40d$ and $N = d = 40$, with a ridge parameter of 1 and variances of $\sigma_1^2 = \sigma_2^2 = 1$. For panel (b), we set $N = d$ while varying both $d$ and $P$. The experimental value of $P_*$ was calculated by extrapolating from the different values of $P$ and their corresponding alignments, and then inverting this correspondence to find the value of $P$ for which $A_f = 0.1$.

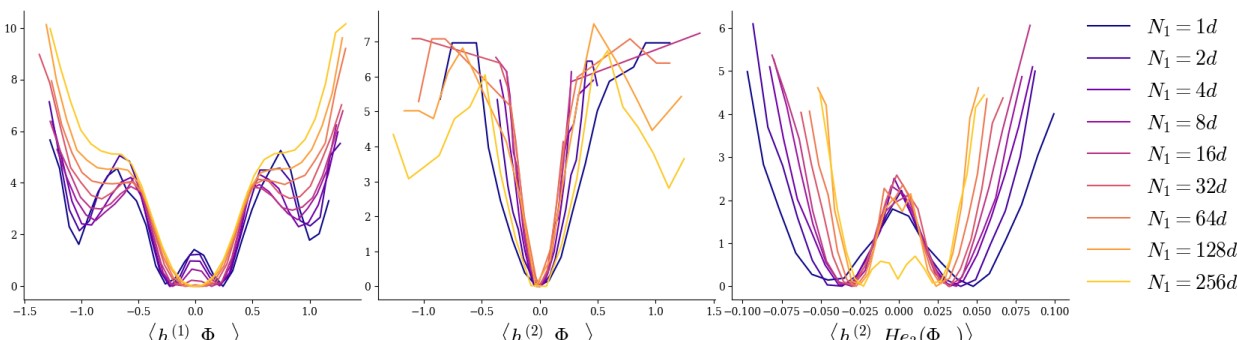

Figure 8: Distribution of pre-activations of the alignment of the first and second layers with the linear feature-$\Phi_* = w_* \cdot x$, and the cubic feature- $\hat{H}e_3(\Phi_*)$ where the hat simply denotes the normalized third Hermite polynomial. As can be seen in this figure, increasing the width of the first layer-$N_1$ pushes the preferred feature learning pattern, from Specialization-magnetization to GP-Specialization, as predicted by the our approach.

In panel (c), we set $P = 40d$ and $d = 40$, while increasing the value of $N$. As in the other panels, we used $\sigma_1^2 = \sigma_2^2 = 1$, but here we set the ridge parameter to $\kappa = 0.25$. We trained an ensemble of 300 networks and computed a histogram of the pre-activation alignment in the hidden layer. First-layer outliers were defined as any pre-activations having an overlap with $\Phi_*(x) = w_* \cdot x$ of more than $3\sigma_1/\sqrt{d}$ (three times the standard deviation of the GP distribution). The total number of such outliers was counted and then averaged over the ensemble of networks.

### F.1 THREE-LAYER FCN- EXPERIMENTAL DETAILS

In panel (a) of Fig. 3, the experimental data was computed by training an ensemble of 30 networks. For all experiments, we set $d = N_1 = N_2$, with a ridge parameter of 0.125 and variances of $\sigma_1^2 = \sigma_2^2 = 0.25$.

For panel (c) of Fig. 3 and in Fig. 8, we set $P = 40d$ and $d = N_2 = 10$, while increasing the value of $N_1$. We further used $\sigma_1^2 = \sigma_2^2 = 0.25$ and a ridge parameter of $\kappa = 0.125$. An ensemble of 300 networks was trained to compute the following histograms for the pre-activation alignments: 1. of the linear feature in the first hidden layer, 2. with the linear feature in the second layer, and 3. with the target in the second layer. Outliers in the first layer were calculated as in the two-layer case. For the second layer, activation alignments deviating by more than 3 times the NNGP standard deviation were considered outliers.

### F.2 ALIGNMENT AS A TIGHT BOUND ON MSE

A central component of our approach is using alignment as a certificate for good learning. As discussed in the main body of the text, the alignment is indeed a lower bound on the MSE through the inequality MSE$\geq (1 - A_f)^2$. However, this inequality does not necessitate that the MSE will vanish. We therefore provide in this section empirical evidence that strong alignment is indeed an indication of MSE. To this extent, we provide results both from the softmax attention experiments and the three layer case, and in both we find that strong alignment is closely correlated with low loss, as can be seen in Fig.

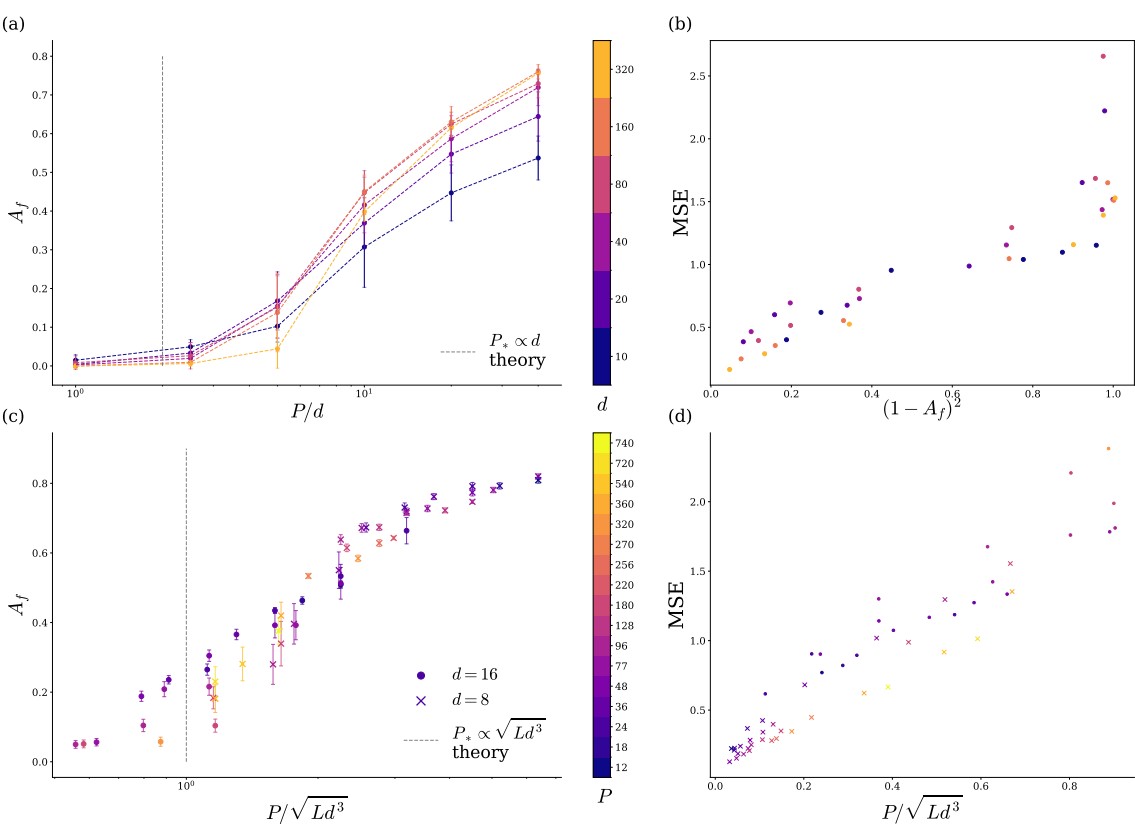

Figure 9: Evidence that alignment is a tight lower bound on MSE. Here we provide results from two experiments, softmax attention (panels(c) and (d)), and three layer network (panels (a) and (b)). These results clearly imply that the alignment is a strong indicator of good MSE.

## G  COMPARING KERNEL ADAPTATION AND LDT

The results in the heuristic approach presented here are closely related to the kernel adaptation approach Fischer et al. (2024); Seroussi et al. (2023c); Ringel et al. (2025); Rubin et al. (2024; 2025). In this section, we compute explicitly the mean posterior predictor for the alignment. Interestingly, we find that the equations for the LDT approach are remarkably similar to those of the mean predictor.

As in the derivation of the Chernhof bounds we begin by integrating out the readout weights from the posterior, and introducing a delta function Fourier field in the prior, we obtain-

$$Z = \int d\tilde{t} \int d\Theta^{L-1} p\left(\Theta^{L-1}\right) \exp\left(-\frac{\sigma_L^2}{2N_L\chi}\sum_{i=1}^{N_L}\left\langle \sigma\left(h_i^{L-1}(x)\right), \tilde{t}(x)\right\rangle^2 - \frac{\kappa}{2}\left\langle\tilde{t},\tilde{t}\right\rangle + i\left\langle y, \tilde{t}\right\rangle\right) \quad (91)$$

where $\tilde{t}$ is an imaginary helper field originating from the delta function. Define-

$$\tilde{t}(x) = t_y y(x) + t_\perp y_\perp(x) \quad (92)$$

where $y_\perp$ is some function that is orthogonal to $y$, so that $\langle y, y_\perp\rangle = 0$. Now we have-

$$Z = \int dt_y dt_\perp \exp\left(-\frac{\kappa}{2}\left(t_y^2 + t_\perp^2\right) + it_y\right)\int d\Theta^{L-1} p\left(\Theta^{L-1}\right) \quad (93)$$

$$\cdot \exp\left(-\sum_{i=1}^{N_L}\frac{\sigma_L^2}{2N_L}\left(t_y\left\langle y(x), \sigma\left(h_i^{L-1}(x)\right)\right\rangle + t_\perp\left\langle y_\perp(x), \phi\left(w_i\cdot x\right)\right\rangle\right)^2\right)$$

Next, we assume self consistently that $t_\perp$ vanishes, and so the above equation simplifies to:

$$Z = \int dt_y dt_\perp \exp\left(-\frac{\kappa}{2}\left(t_y - \frac{i}{\kappa}\right)^2\right)\int d\Theta^{L-1} p\left(\Theta^{L-1}\right)\exp\left(-\sum_{i=1}^{N_L}\frac{\sigma_L^2}{2N_L}t_y^2\underbrace{\left\langle y(x), \sigma\left(h_i^{L-1}(x)\right)\right\rangle^2}_{a_i(\Theta^{L-1})}\right) \quad (94)$$

Here enters the adaptive approach. We assume that $t_y$ is weakly fluctuating, and approximate-

$$\exp\left(\frac{\sigma_a^2}{2N}t_y^2 A_i^2\right) \approx \exp\left(-\sum_{i=1}^{N_L}\frac{\sigma_L^2}{2N_L}\mathbb{E}_{t_y}\left[t_y^2\right]a_i^2\left(\Theta^{L-1}\right) + \sum_{i=1}^{N_L}\frac{\sigma_L^2}{2N_L}t_y^2\mathbb{E}_{\Theta^{L-1}}\left[a_i^2\right]\right) \quad (95)$$

So that

$$Z \propto \int dt_y \exp\left(\underbrace{-\frac{\kappa}{2}\left(t_y - \frac{i}{\kappa}\right)^2 - \sum_{i=1}^{N_L}\frac{\sigma_L^2}{2N_L}t_y^2\mathbb{E}_{\Theta^{L-1}}\left[a_i^2\right]}_{:=-S_{t_y}}\right) \quad (96)$$

$$\cdot\int d\Theta^{L-1}\exp\left(\underbrace{-\sum_{i=1}^{N_L}\frac{\sigma_L^2}{2N_L}\mathbb{E}_{t_y}\left[t_y^2\right]a_i^2\left(\Theta^{L-1}\right) + \log p\left(\Theta^{L-1}\right)}_{:=-S_{\Theta^{L-1}}}\right)$$

We require self consistently that $\mathbb{E}_{\Theta^{L-1}}\left[a_i\left(\Theta^{L-1}\right)\right]$ is indeed given by the expectation with respect to the action $S_{\Theta^{L-1}}$

$$\mathbb{E}_{\Theta^{L-1}}\left[a_j\left(\Theta^{L-1}\right)\right] = \mathbb{E}_{\Theta^{L-1}\sim S_{\Theta^{L-1}}}\left[a_j\left(\Theta^{L-1}\right)\right] \tag{97}$$

$$= \frac{\int d\Theta^{L-1}p\left(\Theta^{L-1}\right)a_j\left(\Theta^{L-1}\right)\exp\left(-\frac{\sigma_L^2}{2N_L}\mathbb{E}_{t_y}\left[t_y^2\right]\sum_{i=1}^{N_L}a_i\left(\Theta^{L-1}\right)\right)}{\int d\Theta^{L-1}p\left(\Theta^{L-1}\right)\exp\left(-\sum_{i=1}^{N_L}\frac{\sigma_L^2}{2N_L}\mathbb{E}_{t_y}\left[t_y^2\right]\sum_{i=1}^{N_L}a_i\left(\Theta^{L-1}\right)\right)}$$

Denoting- $\sum_{i=1}^{N_L-1}a_i\left(\Theta^{L-1}\right) := a\left(\Theta^{L-1}\right)$, and $\int d\Theta^{L-1}p\left(\Theta^{L-1}\right)(...) = \mathbb{E}_{\Theta^{L-1}\sim\text{GP}}\left[(...)\right]$, then we have-

$$\sum_{j=1}^{N_L}\mathbb{E}_{\Theta^{L-1}\sim S_{\Theta^{L-1}}}\left[a_j\left(\Theta^{L-1}\right)\right] := \mathbb{E}_{\Theta^{L-1}\sim S_{\Theta^{L-1}}}\left[a\left(\Theta^{L-1}\right)\right] \tag{98}$$

$$= \frac{\mathbb{E}_{\Theta^{L-1}\sim\text{GP}}\left[a\left(\Theta^{L-1}\right)\exp\left(-\frac{\sigma_L^2}{2N_L}\mathbb{E}_{t_y}\left[t_y^2\right]a\left(\Theta^{L-1}\right)\right)\right]}{\mathbb{E}_{\Theta^{L-1}\sim\text{GP}}\left[\exp\left(-\frac{\sigma_a^2}{2N}\bar{t}_y^2 a\left(\Theta^{L-1}\right)\right)\right]}$$

Next we require that the same condition holds for $t_y$. Note that the action for $t_y$ is simply a Gaussian one, and following square completion we have-

$$e^{-S_{t_y}} \propto \exp\left(-\frac{\left(\kappa + \frac{\sigma_L^2}{N_L}\mathbb{E}_{\Theta^{L-1}}\left[a\left(\Theta^{L-1}\right)\right]\right)}{2}\left(t_y - \frac{i}{\left(\kappa + \frac{\sigma_L^2}{N_L}\mathbb{E}_{\Theta^{L-1}}\left[a\left(\Theta^{L-1}\right)\right]\right)}\right)^2\right) \tag{99}$$

So that

$$t_y \sim \mathcal{N}\left(i\left(\kappa + \frac{\sigma_L^2}{N_L}\mathbb{E}_{\Theta^{L-1}}\left[a\left(\Theta^{L-1}\right)\right]\right)^{-1}, \left(\kappa + \frac{\sigma_L^2}{N_L}\mathbb{E}_{\Theta^{L-1}}\left[a\left(\Theta^{L-1}\right)\right]\right)^{-1}\right) \tag{100}$$

So that we can substitute the self consistency requirement on the $\Theta$ variables into the expression for the mean of $t_y$ and we obtain

$$\mathbb{E}_{t_y\sim S_{t_y}}\left[t_y\right] = i\left(\kappa + \frac{\frac{\sigma_L^2}{N_L}\mathbb{E}_{\Theta^{L-1}\sim\text{GP}}\left[a\left(\Theta^{L-1}\right)\exp\left(-\frac{\sigma_L^2}{2N_L}\mathbb{E}_{t_y\sim S_{t_y}}\left[t_y^2\right]a\left(\Theta^{L-1}\right)\right)\right]}{\mathbb{E}_{\Theta^{L-1}\sim\text{GP}}\left[\exp\left(-\frac{\sigma_a^2}{2N}\mathbb{E}_{t_y\sim S_{t_y}}a\left(\Theta^{L-1}\right)\right)\right]}\right)^{-1}$$

Note that if we approximate- $\mathbb{E}_{t_y\sim S_{t_y}}\left[t_y^2\right] \sim \mathbb{E}_{t_y\sim S_{t_y}}^2\left[t_y\right]$, then we obtain a self consistent equation for $\mathbb{E}_{t_y\sim S_{t_y}}\left[t_y\right] := \bar{t}_y$, given by-

$$\bar{t}_y = \frac{i\mathbb{E}_{\Theta^{L-1}\sim\text{GP}}\left[\exp\left(-\frac{\sigma_L^2}{2N_L}\bar{t}_y^2 a\left(\Theta^{L-1}\right)\right)\right]}{\mathbb{E}_{\Theta^{L-1}\sim\text{GP}}\left[\left(\kappa + \frac{\sigma_L^2}{N_L}a\left(\Theta^{L-1}\right)\right)\exp\left(-\frac{\sigma_L^2}{2N_L}\bar{t}_y^2 a\left(\Theta^{L-1}\right)\right)\right]} \tag{101}$$

Rewriting with real $\bar{t}_y \mapsto i\bar{t}_y$, we obtain the self consistency equation-

$$\bar{t}_y = \frac{\mathbb{E}_{\Theta^{L-1}\sim\text{GP}}\left[\exp\left(\frac{\sigma_L^2}{2N_L}\bar{t}_y^2 a\left(\Theta^{L-1}\right)\right)\right]}{\mathbb{E}_{\Theta^{L-1}\sim\text{GP}}\left[\left(\kappa + \frac{\sigma_L^2}{N_L}a\left(\Theta^{L-1}\right)\right)\exp\left(\frac{\sigma_L^2}{2N_L}\bar{t}_y^2 a\left(\Theta^{L-1}\right)\right)\right]} \tag{102}$$

Replacing $\tilde{a} = \kappa + \frac{\sigma_L^2}{N_L} a\left(\Theta^{L-1}\right)$, we obtain the following self consistent equation for $\bar{t}_y$:

$$\bar{t}_y = \frac{\mathbb{E}_{\Theta^{L-1}\sim\mathrm{GP}}\left[\exp\left(\frac{\sigma_L^2}{2N_L}\bar{t}_y^2\tilde{a}\left(\Theta^{L-1}\right)\right)\right]}{\mathbb{E}_{\Theta^{L-1}\sim\mathrm{GP}}\left[\left(\frac{\sigma_L^2}{N_L}\tilde{a}\left(\Theta^{L-1}\right)\right)\exp\left(\frac{\sigma_L^2}{2N_L}\bar{t}_y^2\tilde{a}\left(\Theta^{L-1}\right)\right)\right]} \tag{103}$$

This expression is very similar to the one we found for the upper Chernoff bound, and the two equations coincide in the limit $\kappa \to 0$, and $\alpha \to 1$.