# OpenReview forum: "Mitigating the Curse of Detail: Scaling Arguments for Feature Learning and Sample Complexity"
_ICLR.cc/2026/Conference — ICLR 2026 Poster_

### Official Review · Reviewer_d7wZ · 2025-10-27

**Soundness:** 4
**Presentation:** 2
**Contribution:** 3
**Rating:** 6
**Confidence:** 3

**Summary:**

This paper proposes a new method to estimate sample complexity, i.e., the amount of data needed in order to learn a target function (measured in terms of alignment). The authors propose a bound in function space depending on how well a randomly drawn NN would align with the target. As this bound is uncomputable, they derive a variational approach based on a mean-field decoupling. They hypothesize two feature learning mechanisms that go beyond the NNGP baseline and derive certain signatures of feature learning, like eigenvalue spikes in the conjugate kernel and a description of feature learning in a three-layer model. This is verified on one empirical target function.

**Strengths:**

- Clear quantitative predictions for general architectures which go way beyond the usual toy models or qualitative results.

- Very general framework that can be applied independently of architecture, data distribution, etc.

- New analysis of a three-layer non-linear network.

**Weaknesses:**

- Hard to follow at some places, especially the three-layer NN section. Lines 282-301 are very dense and should be made clearer in the camera-ready version.

- The theory needs much more empirical validation. I.e., what happens with other target functions, other input distributions, etc.? Are the predictions still correct?

- How do we know the proposed distributions actually minimize the variational energy? The paper would greatly benefit from a comparison of an SGLD-trained network and how close the neuron distribution of this network is to the ones proposed as the minimizers of the variational energy.

- Some typos: line 70 misses a "." after the eq., lines 70 and 81 miss a "," after the eq., line 84 $x_{\nu}$ is not the dataset but a point, line 219 has a "," at the end instead of a ".", $\Theta$ is not defined.

- The central idea of the paper is related to [1] and especially to [2] where analytical bounds on the prior are derived in a simplified setting, but this work is not cited.

[1]: https://arxiv.org/abs/2006.15191
[2]: https://arxiv.org/abs/2505.24060

**Questions:**

- How does a bound on the prior correctly imply sample complexity for a feature learning network? Even after reading the appendix, I do not fully understand how this is possible.

- How does the structure of the target function influence the sample complexity in this framework? I.e., if the target function only depends on $k\ll d$ input coordinates, can it be shown that the sample complexity of the NN in that framework scales with $k$ not $d$ more generally?

- How is $P_{*}$ depending on the input distribution of the data?

- How do we know that the proposed minimizers of the variational densities NNGP, GFL and specialization are the true minimizers?

- How does this connect to PAC-Bayes bounds? (https://arxiv.org/pdf/2110.11216)

---

> ### Author Response · Authors · 2025-11-26
> **Response to reviewer d7wZ: Part 1**
>
> We thank Reviewer d7wZ for their valuable feedback and positive
> response. Following careful consideration, we made multiple changes to
> the main text to address the relevant points made by the reviewer.
>
> -   Improvements to presentation: We made significant edits,
>     including: (1) Fixing the typos mentioned by this reviewer and
>     others, and clarifying definitions. (2) Adding a schematic overview
>     of our proposed framework. (3) Including an explicit summary of
>     kernel propagation rules, which we then used directly in the
>     computation of concrete cases. (4) Including a much expanded
>     description of how the variational approximation was obtained in the
>     case of a three-layer network. (5) An additional overview paragraph
>     in the introduction explaining the logic flows.
>
> -   Extensions to empirical validation: In the revised version, we
>     included analysis of a transformer, as well as a ReLU network
>     performing a sparse parity classification tasks. The latter reveals
>     a qualitatively different behaviour than that of a bounded
>     activation such as erf, correctly captured by our framework. In
>     total are heuristic was verified in four differentiated settings and
>     explains known experimental results ($P=d^{3/4}$ in CNNs).
>
> -   Clarification of experimental validation: We indeed compare our
>     theoretical estimate both to an SGLD-trained network, and, for the
>     two-layer FCN, we compare it to the exact analytical solution. The
>     theoretical curves for two-layer networks were mistakenly left
>     without a label, an omission which has been corrected in the
>     revision. We also find good agreement both in the form of the
>     distribution and in quantities such as the number of specializing
>     neurons in the three-layer networks, and demonstrate this explicitly
>     in our revision.
>
>     Despite the good agreement presented in our work, we cannot
>     guarantee that these indeed minimize the variational energy. The
>     goal of proposing these candidate distributions was to enable
>     simplified calculations, and thus we reduce the variational ansatz
>     to a finite number of candidate distributions. Since these patterns
>     that we have selected closely correspond to observed ones, we make
>     the reasonable assumption that one of them will indeed minimize the
>     variational energy.
>
> -   Included missing citations: We thank the reviewer for pointing out
>     these important and relevant citations and included them in the
>     revised main text.
>
> In response to the reviewer's questions:
>
> -   Relation between prior bound and sample complexity: We agree that
>     this is a counterintuitive point. First, we emphasize that the upper
>     bound is in fact on the posterior but is set by the prior and the
>     dataset size. The general idea, proven in App. A, is that an outcome
>     that is extremely unlikely according to the prior needs a lot of
>     data to become a favourable explanation in the posterior. Loosely,
>     it's like the common saying that extraordinary claims (prior
>     unlikely) require extraordinary evidence (many data points). The
>     most extreme case would be the fact that if an outcome (e.g.
>     alignment of $1$) has zero chance according to the prior, it also
>     has zero chance to appear in the posterior for any finite $P$ based
>     on the formula for the posterior.
>
>     Even so, it may feel miraculous that the prior can encode feature
>     learning effects. The way this comes about technically is by the
>     following route. Having strong alignment in a network drawn from the
>     prior is a rare event, and rare events are often driven by entirely
>     different random mechanisms than typical events, the latter being
>     more Gaussian/GP-like. The standard way of analyzing rare events,
>     'tilts' the prior by a factor which makes the rare event likely (see
>     first equation in App. Sec. B.2.). That tilting, on a qualitative
>     level, plays the role of the likelihood term in the posterior. Thus,
>     **conditioned** on a rare-event, random networks are closer to
>     posteriors than to priors.
>
> -   Generalization to multi-index models: Generally, we expect that
>     $\tilde{E}\_q$ will be extensive in the number of features. For
>     instance, if our target is
>     $y(x) = \sum\_{c=1}^C a^{\*}\_c ReLU(w^{\*}\_c \cdot x)$, we'd expect $C$
>     specializing neurons, yielding a $dC$ cost in $\tilde{E}\_q$. Our new
>     classification setting partially addresses such scenarios as the
>     networks needs to specialized to more than one feature, although we
>     find that it is does not significantly affect our theory in this
>     particular setting.

---

> > ### Author Response · Authors · 2025-11-26
> > **Response to reviewer d7wZ: Part 2**
> >
> > -   Sample complexity and input distribution of the data: Since our
> >     framework focuses on the prior, much of it is agnostic to the input
> >     data and allows working with any convenient data measure. However, a
> >     dependence can nevertheless emerge through the measure on which the
> >     alignment is defined. The choice of data measure can be arbitrary in
> >     principle, but a reasonable practical choice to make is to take the
> >     same measure as that from which the input data is drawn. However,
> >     other options can also be desirable, and we view this freedom as a
> >     key advantage of our approach, as it allows us to study
> >     out-of-distribution generalization capabilities and provide explicit
> >     results even when the distribution is complex or unknown (e.g.
> >     real-world datasets). For instance, consider a network trained to
> >     learn parity, but only trained on data where all bits are the same
> >     up to some small random noise. Taking the training data as the
> >     alignment measure would seemingly result in perfect generalization
> >     with very few samples, because the constant function approximates
> >     parity very well, apart from when the noise kicks in. Taking other
> >     data distributions for computing the alignment (e.g. random bits)
> >     would then reveal that the network did not, in fact, learn the
> >     underlying parity problem.
> >
> > -   Minimization condition: In this work, we do not attempt to guarantee
> >     that these patterns will indeed minimize the distribution. Better
> >     patterns may exist, though, based on existing literature, the ones
> >     contained in this work are a significant sample. We show that the
> >     predicted patterns indeed emerge in the case of a two and
> >     three-layer nonlinear network in our work, as well as in a two-layer
> >     ReLU network.
> >
> > -   Relation to PAC-Bayes bounds: There are various interesting
> >     parallels (as well as differences) between PAC-Bayes bounds and our
> >     approach. **(i)** PAC-Bayes bounds provide upper bounds on sample
> >     complexity, namely, they (probabilistically) ensure learning above a
> >     threshold $P>P_*$ at given accuracy. Our bounds ensure no learning
> >     for $P < P_*$. In other words, we focus on establishing limitations
> >     of deep learning architectures, which, we believe, is as or more
> >     interesting than rationalizing their success. **(ii)** The PAC
> >     bounds often involve the KL-divergence between prior and posterior
> >     ($D_{KL}(\rho|| \pi)$, notably, $\rho$ being the posterior here and
> >     $\pi$ the prior), plus some log terms, over the sample size.
> >     Ensuring this term is small, requires $P > D_{KL}(\rho|| \pi)$. In
> >     fact, ignoring the relatively inconsequential logarithmic terms and
> >     making the reasonable assumption that the empirical risk (training
> >     loss) is also small, having $P \gg D_{KL}(\rho|| \pi)$ ensures
> >     learning (see for instance, McAllester, D. A. (1999). "PAC-Bayesian
> >     model averaging"). A key point is that, via the
> >     data-processing-inequality, if we define a Bernoulli variable for
> >     the $A_f > \alpha$ event in the posterior and prior (${\rm P}$ and
> >     $\Pi$ respectively) then
> >     $D_{KL}(\rho|| \pi) \ge D_{KL}({\rm P}||\Pi)$. Further taking the
> >     posterior to have
> >     $\Pr\nolimits_{\rho}\left[A_{f}\geq\alpha\right] = 1$, we obtain a
> >     lower-bound on the $P$ needed for good learning namely
> >     $P > D_{KL}(\rho|| \pi) \ge D_{KL}({\rm P}|\Pi) = -\log(\Pr\nolimits_{\pi}\left[A_{f}\geq\alpha\right])$.
> >     This coincides with our prior bound.

---

> > ### Author Response · Authors · 2025-11-27
> > **Clarification of experimental extensions**
> >
> > Regarding the extensions to our empirical validation (addressing the second weakness), we emphasize that the classification analysis included in the revision introduces an additional data distribution. Specifically, we consider input data that is drawn from a bimodal Gaussian mixture with means $\pm 1$.

---

### Official Review · Reviewer_9jc9 · 2025-10-30

**Soundness:** 2
**Presentation:** 2
**Contribution:** 3
**Rating:** 4
**Confidence:** 2

**Summary:**

The paper proposes a scaling-based recipe for predicting when and where specific feature learning attributes (covariance spikes, neuron specialization) appear in deep nets. It uses a Bayesian description of neural networks to analyze the sample-size threshold needed to have $O(1)$ alignment  between a function (a kernel eigenmode) and the network. Heuristic propagation rules connect layers (e.g., specialization induces a downstream spectral spike; spikes amplify higher-order features), letting predictions compose across depth. Worked cases for data scaling exponents are given for 2/3-HL FCNs, as well as CNNs.

**Strengths:**

There appears to be quality work done on the mathematics behind this paper’s analysis, as well as an original approach to mech interp. It’s clear not enough similar work is done in that community, so the authors should be applauded for their instinct to pursue this direction. Another major strength is connecting the results obtained back to previous findings, this time under a new light. The main math-heavy sections are well presented. This work appears to be quite significant to those in the mech interp field willing to read through the paper to understand its ideas and implications.

**Weaknesses:**

This paper would greatly benefit from plots showing how the theory matches results, not just empirical results which are quite difficult to connect back to specifically this work and not some other broad notions of the field. Another weakness is in the downplay of grokking: since this theory is entirely about near-convergence networks, it seems like there should be a much greater focus on things like weight decay or grokking; outside of the intro, I don’t see those mentioned again. Finally, some things need to be defined instead of assumed (take Figure 1(a)’s y-axis, what is H_pi?)

**Questions:**

Why is the error function being used for the activation function? If any can be used, why not just the standard ReLU?
Can you clarify the point of the claims in 4.4?
Can plots of the ConvNet case be provided? In other words, are the details you’re trying to abstract away the same ones that make ConvNets and MLPs appear differently? If so, why perform this analysis? Those differences seem key to understanding ConvNets.
This is framed as a new framework that is easier to apply than other approaches, so why isn’t it applied to make new predictions? Show me this goes beyond specific cases like 3 layer MLPs on isotropic data with the (seemingly specific) erf activation function

---

> ### Author Response · Authors · 2025-11-26
> **Response to reviewer 9jc9: Part 1**
>
> We thank the reviewer for their feedback, and for recognizing the potential contribution of this work. Regarding weaknesses raised:
>
> -   Relating theory and experiment: The theoretical curves for two-layer
>     networks were mistakenly left without a label. This has been
>     corrected in the revision, and the definition of the $y$ axis of
>     this figure has been clarified as well. We also included in the
>     revision our theoretically predicted scaling curves which match
>     experiments. We further provide several new experiments, which match
>     our scaling predictions, for an attention block, as well as a ReLU
>     network trained on a classification task.
>
> -   Downplay of grokking and weight decay: We argue that our work
>     provides an explanation for grokking, though from a non-canonical
>     equilibrium rather than a dynamical perspective. In Power et al.'s
>     seminal work, grokking was demonstrated as an equilibrium effect
>     driven by training data fraction. Our view matches Power's and more
>     specifically Rubin et al. (2024), where grokking is essentially a
>     1st order phase transition in the learned representation. We find
>     that the emergence of specialization in our model closely
>     corresponds to the phase transition identified in that work (e.g.
>     our Fig. 2a of this work compared to Fig. 2b of Rubin et. al.). In
>     fact, we conjecture that whenever the variational energy of a GFL
>     scenario is inferior to that of a specialization scenario, a first
>     order phase transition would occur in the equilibrium problem,
>     leading to grokking-like behavior in the dynamics. Nevertheless,
>     because the relationship between dynamics and grokking is not an
>     established theory yet, but rather an empirical observation with a
>     plausible physics analogy, we did not delve into this in the text.
>     The revised version now briefly mentions this connection when we
>     introduce the specialization pattern.
>
>     Turning to the topic of weight decay, we are not entirely sure what
>     the reviewer has in mind. In particular, all of our networks are
>     trained with some (typically small) weight decay to allow for a
>     well-defined equilibrium distribution in the presence of Langevin
>     gradient noise.

---

> > ### Author Response · Authors · 2025-11-26
> > **Response to reviewer 9jc9: Part 2**
> >
> > In response to the reviewer's questions:
> >
> > -   Choice of activation: For the explicit solution in the case of a
> >     two-layer network, the error function activation is a natural choice
> >     for analytical tractability. We also included in the revision a
> >     softmax activation transformer. This formalism can be similarly
> >     applied to ReLU, and we included treatment of this activation in a
> >     classification setting the revised version. We find that here too
> >     our heuristics can be applied, and we are able to make concrete
> >     predictions on the scaling of the feature learning magnitude.
> >
> > -   Claims in 4.4: These claims are designed to aid the calculation of
> >     the propagation of the feature learning effects through the kernel
> >     of a deep network. In the expression for the variational energy we
> >     obtain, each layer depends on the pattern of the previous layer
> >     through the kernel. The exact manner in which these effects emerge
> >     requires complex computation. In section 4.4 (5.2 in the revised
> >     manuscript), we provide heuristics for how to compute the feature
> >     propagation. In the revision, we also included the explicit
> >     resulting rules which were used to make the calculations for the
> >     concrete cases in the following section.
> >
> > -   ConvNet results: \"*Can plots of the ConvNet case be provided?*\" We
> >     are referring in our work to previously published results that
> >     capture this scaling behaviour, both analytically, and
> >     experimentally. For the experimental results, see the following:
> >     https://arxiv.org/pdf/2502.18553, specifically Fig. 3.2 on page 96,
> >     and Fig. 3.3 on page 99\
> >     \"*Are the details you're trying to abstract away the same ones that
> >     make ConvNets and MLPs appear differently?*\"\
> >     Our theory predicts a fundamental difference between MLPs and
> >     ConvNets. Thanks to the weight sharing of these networks, the sample
> >     complexity is reduced from $d$ (as is the case for an MLP or an
> >     extremely overparametrized CNN) to $d^{3/4}$. The abstracted details
> >     are not related to the particular architecture of the MLP, the focus
> >     on this simple architecture was only for the sake of clarity in the
> >     presentation.
> >
> > -   Framework novelty: In the revised version, we extended our analysis
> >     to to include what we believe is novel discovery. We also believe
> >     that ability to capture feature learning in such vastly different
> >     settings, from convolutional networks to attention heads is a
> >     significant contribution in its own right. Nevertheless, we would
> >     also like to point out that the predictions for a three-layer
> >     network are indeed new.\
> >     Another important distinction between our work and existing
> >     literature is our ability to predict scaling behaviour. Although
> >     other works have analyzed similar systems, due to the complex
> >     high-dimensional nature of the solution, definitive statements about
> >     sample complexity scaling could not be determined directly. Instead,
> >     numerical solvers for the analytical equations had to be employed on
> >     a case-by-case basis. The ability to determine the scale in a
> >     transparent manner is, therefore, a new result as well. This
> >     strength of our work is especially evident for the understanding of
> >     feature learning, as our work is the first to predict how these
> >     effects scale rather than employing a setting-specific numeric
> >     analysis.

---

### Official Review · Reviewer_tHzt · 2025-11-01

**Soundness:** 1
**Presentation:** 1
**Contribution:** 2
**Rating:** 4
**Confidence:** 3

**Summary:**

This paper proposes a theoretical framework for understanding feature learning through an alignment-based measure that connects learned representations with test mean squared error (MSE). The authors derive inequalities suggesting that when alignment is close to one, the model generalizes well, and they further develop probabilistic bounds on alignment exceeding certain thresholds. The motivation is strong: linking alignment to generalization performance could offer a compact diagnostic tool for feature learning across architectures.

However, I found the mathematical exposition—particularly in Sections 2, 3, and 4—very confusing and difficult to follow. The direction of key inequalities, the interpretation of constants, and the definitions of several introduced symbols are unclear. As a result, I cannot confidently assess the soundness of the theoretical claims, even though the overall goal of the paper is promising.

The paper presents an interesting and well-motivated idea, but the core theoretical exposition is too opaque to evaluate. The logical direction of key inequalities and the meaning of several symbols are unclear, which makes the central claims about alignment as a generalization proxy hard to verify. With clearer derivations, well-defined notation, and a complementary upper bound relating alignment to test MSE, this work could become significantly more compelling.

**Strengths:**

* Strong motivation and relevance: The paper tackles an important and timely question—how to quantify representation quality via alignment—and aims to connect it to generalization performance.
* Potential for theoretical contribution: If clarified, the proposed framework could bridge geometric and probabilistic views of feature learning, an area of broad interest to both machine learning and neuroscience communities.
* Clear high-level narrative: The intuition that alignment captures “how well learned features match task structure” is appealing and consistent with empirical trends observed in deep learning.

**Weaknesses:**

* Unclear and inconsistent mathematical exposition: The logic behind the central inequality (around line 98) is confusing. It provides only a lower bound on test MSE in terms of alignment, so high alignment is at best a necessary—but not sufficient—condition for good generalization. The paper does not establish or even discuss whether an upper bound exists, which weakens the claim that alignment close to 1 can reliably indicate low test error.
* Ambiguity in probabilistic statements: In Section 3, the analysis of “probability of alignment > α” is difficult to interpret because the relevant range of α is not specified and the meaning of O(1)  is unclear—does it denote “around one” or simply “a constant independent of dimension”? This ambiguity propagates through subsequent results.
* Notation and definition issues: Several key quantities (e.g., H_{p,α}, q, and related expressions around lines 146–152) are either undefined or only referenced indirectly via the appendix. The paper should clearly define each symbol in the main text, especially when these terms appear in central inequalities.
* Direction of inequalities unclear: In Section 4, it is unclear whether P(A)F(α)) is meant to upper bound or lower bound the probability of alignment exceeding α. The sign and logical direction of several inequalities seem inconsistent, making it hard to reconstruct the intended argument.
* Difficult to evaluate validity: Because of these ambiguities, it is impossible to judge whether the mathematical derivations are correct or merely stated heuristically. The empirical illustrations in later sections cannot compensate for this lack of clarity in the core theory.

**Questions:**

1. In line 98, how does the lower bound on test MSE justify using alignment as a reliable proxy? Is there also an upper bound, or an argument showing that alignment close to 1 implies low error in practice?
2. In Section 3, what is the intended range for α, and what does O(1) mean precisely in this context?
3. In Section 4, should PAF(α) serve as an upper or lower bound on the probability of high alignment? Please clarify the direction and logic of this inequality.
4. Please define all symbols that appear in main derivations (e.g., H_{p,α}, q, etc.) in the text rather than referring only to the appendix.
5. Could you include a clear schematic or conceptual figure illustrating how alignment links to generalization, and summarize the assumptions under which the inequalities hold?

---

> ### Author Response · Authors · 2025-11-26
> **Response to reviewer tHzt: Part 1**
>
> We thank the reviewer for their thorough review. We have carefully
> considered each point raised and made significant revisions which we
> hope will help clarify our intentions. In addition to the technical
> issues discussed below, we would like to point out that (1) we conducted
> two more experiments that validate our heuristic approach (softmax
> attention block and sparse parity classification task) and (2) our work
> explores a theoretically motivated heuristic, as is common in
> theoretical studies of complex physical systems, rather than aim for a
> closed form prediction at full generality.\
> Following Reviewer tHzt feedback, we made the following amendments:
>
> -   Clarification of the assumptions on the relation between MSE and
>     alignment: as the reviewer correctly pointed out, our work
>     establishes a necessary, not a sufficient, condition for learning.
>     Indeed, we establish an upper bound on the alignment and, thus, a
>     lower bound on test-MSE. While establishing a complementary bound is
>     left for future work, we provide both evidence for its tightness, as
>     well as theoretical motivation for considering this quantity as a
>     learning certificate over MSE. Specifically, we find empirically
>     that alignment is often a strong indicator of low MSE. Following
>     this reviewer's suggestion, we included in the appendix a comparison
>     of alignment vs. MSE, which demonstrates that the MSE satisfies
>     empirically MSE $\propto (A_f-1)^2$ in both FCN.
>
>     Finally, we note that, given the transformer's incredible abilities
>     across so many tasks and domains, what networks *can't* learn (i.e.
>     lower bounds on $P$) is as interesting a question these days as what
>     they can learn (i.e. upper bounds).\
>
> -   Corrected ambiguity in probabilistic statements: we appreciate the
>     reviewer for pointing out this ambiguity in our notation. We revised
>     our notation, replacing $\alpha \sim \mathcal{O}(1)$ with
>     $\alpha \approx 1$. While $\alpha$ may attain any real value, the
>     majority of our analysis and approximations hold for
>     $\alpha \approx 1$. We also revised our manuscript to better stress
>     where we make the assumption on $\alpha$.
>
> -   Clarification of notation and definition: following the reviewer's
>     comment, we closely reviewed the definitions and notations used in
>     this work to clarify definitions of the following quantities:
>
>     -   Definition of $H_{p,\alpha}(h)$: Equation (6) served as a
>         definition for $H_{p,\alpha}(h)$ derived from the previously
>         defined $p_{A_f}(\alpha)$, namely, for every $\alpha$-density
>         one can find a Hamiltonian $H_{p,\alpha}(h)$ and a normalizing
>         constant $Z_p$ such that (6) holds (eq (7) in the revised
>         version). However, this definition is not unique, since we would
>         be free to shift $H$ by a constant, and that would simplify
>         correspond to a different normalization constant. We resolve
>         this ambiguity in the revision by defining $H$ so that
>         $\min_{h}H_{p}(h)=0$. We also explain this notation explicitly.
>
>     -   Definition of $q$: We defined a variational density $\hat{q}(h)$
>         over preactivations, and used $q$ as an index referring to the
>         same distribution. We address this ambiguity by eliminating the
>         use of $\hat{q}$ and strictly using $q$ as a density over $h$.
>         When we write $q(h)=e^{-H_q(h)}/Z_q$, we mean this in the sense
>         of the previous point, namely, there exists $H_q(h)$ and a
>         normalizing constant $Z_q$ establishing a density $q$ over $h$.
>
>     -   Definition of $\tilde{E}\_{q_{\*}}$:We now explicitly define this to
>         be the variational energy for the minimizer $q\_{\*}$.
>
>     -   We clarified the source of $\sigma_l$ in the definition of the
>         kernel
>
> -   Direction of inequalities unclear: we agree that the direction of
>     the inequality is confusing. We clarify this issue as follows (and
>     in the revised main text). In short, we obtain a lower bound on the
>     minimal sample size necessary for learning, $P_{\*}$. Then, a necessary
>     condition for learning is that the sample size
>     $P \geq P_{*\} \propto E(\alpha)$. However, the energy $E(\alpha)$ is
>     challenging to compute directly, and so we turn to variational
>     approximation (specifically the Feynman-Bogoliubov inequality). In
>     principle, this inequality results in an upper bound on $E(\alpha)$
>     in terms of the variational energy: $E(\alpha) \leq \tilde{E}\_{q}$.
>     Following standard arguments from statistical mechanics, we assume
>     the upper bound is a sufficiently close approximation:
>     $E(\alpha) \approx \tilde{E}\_{q}$. To conclude, we establish a
>     necessary condition for learning:
>     $P \geq P_{\*} \propto E(\alpha) \approx \tilde{E}\_{q}$.

---

> > ### Author Response · Authors · 2025-11-26
> > **Response to reviewer tHzt: Part 2**
> >
> > Regarding the reviewer's questions:
> >
> > 1.  See the first bullet point above.
> >
> > 2.  Please see the second bullet above.
> >
> > 3.  The probability density $p_{A_f}(\alpha)$ is used to bound the
> >     minimal sample size necessary for strong alignment rather than the
> >     probability of strong alignment itself. This bound is obtained
> >     through the following expression:
> >     $P_{\*} \propto -\log p_{A_f}(\alpha)$. For large $\alpha$ we can
> >     further approximate $-\log p_{A_f}(\alpha)\approx E(\alpha)$.
> >
> > 4.  See the third bullet above.
> >
> > 5.  We thank the reviewer for the suggestion, and include in the revised
> >     edition a schematic overview of our framework, which includes a summary of
> >     all suggestions. As mentioned previously, we also include a figure
> >     relating MSE to the alignment, and further explain how the alignment
> >     can be used to better understand generalization.

---

### Official Review · Reviewer_eTgY · 2025-11-02

**Soundness:** 3
**Presentation:** 3
**Contribution:** 3
**Rating:** 6
**Confidence:** 3

**Summary:**

The paper uses variational energy bound and proposes a set of heuristics to characterize the scaling property of sample complexity ($P^*$ such that $logPr_{p_0}(A_f \geq \alpha)$ . The heuristics (which is an approximation of the variational energy) are developed for the feature learning patterns of Gaussin Process, Gaussain Feature Learning, and Feature Specialization. The paper then uses these heuristics to estimate the variational energy of these feature learning patterns of Fully-Connected 2-layer networks, CNN with Non-overlapping Patches, and Three-layer Network

**Strengths:**

The paper addresses a well-defined problem, which is how to estimate the scaling property of sample complexity, in deep network, with is usually analytically intractable. To address this problem, the author uses lower-bound variational energy and a set of heuristics.
The paper supports their claim by both theoretical results and simulations, and show that these heuristics can estimate the lower bound variational energy and derive the sample complexity for fully-connected network and CNN with non-overlapping patches, which match with previous analytical results in the literature.
The paper also shows that their method can be applied to a more complicated network such as three-layer neural networks, which is difficult to get analytical results

**Weaknesses:**

As the paper employs heuristics to estimate the the lower variational bound for a specific set of feature learning patterns (Gaussian Process, Gaussian Feature Learning, Feature Specialization), under a specific task setup (polynomial of degree m), it's difficult to evaluate how these heuristics of specific feature learning patterns can be applicable for different tasks (classification, regression with a more general basis functions). I would appreciate if the authors could discuss the expected applicability or limitations of their framework beyond the polynomial setting.

**Questions:**

1. In Eq. 7 and Eq. 8, how does $\tilde{E}_q$ depends on $\alpha$?
2. (Fig 2B) How do we compute the number of specialist neurons in each layer?
3. How does the three feature learning patterns (GP, GFL, and Specialization) get selected? Would these patterns cover most feature learning space?

---

> ### Author Response · Authors · 2025-11-26
> **Response to reviewer eTgY**
>
> We thank the reviewer for their accurate account of our work and their general positive review. Our analysis does apply to a broader range of settings than simply the polynomial one. To demonstrate this more clearly, our revised manuscript includes a wider variety of tasks, such as a softmax attention block as well as classification tasks trained with cross-entropy loss. Notwithstanding, we also point out that the polynomial task is indicative as well, since Hermite polynomials are a spanning basis for $x\sim\mathcal{N}(0,I_d)$.
>
> In response to Reviewer eTgY questions:
> 1. In the computation of the variational energy, $\tilde{E}_q$, we assume that $\alpha\approx 1$. As we are interested in scaling behaviour, the exact value of $\alpha$ is not significant, but for the approximations to hold, we require a finite $\alpha$. As this point was not clear, we revised the notation in our manuscript and now define explicitly when the assumptions on $\alpha$ enter the approximation.
> 2. As explained in the experimental details appendix, outliers were defined as any pre-activations having an overlap with $\Phi_*(x) = w_* \cdot x$ of more than $3\sigma_1/\sqrt{d}$ (three times the standard deviation of the GP distribution). The total number of such outliers was counted and then averaged over an ensemble of 300 networks for each data point in the figure.
> 3. The patterns selected in this work encompass, to the best of our knowledge, all those that have been previously used in theories of rich learning (including Van Meegen & Sompolinski 2024, Fischer et al 2024; Rubin et al 2024; Seroussi et al 2023). Nevertheless, our heuristic framework could accommodate other patterns if deemed relevant. For instance, one may imagine a synchronized/correlated-specialization pattern where several neurons specialize together. However, even if so, once the pattern is understood, the variational energy cost can still be approximated by our $\tilde{E}$. Given that one accepts this relatively small "search-space" of patterns, one should still find the optimal pattern. This is done by considering a plurality of patterns, and selecting the one with the best $\tilde{E}_q$. This process, for the particular case of a 3-layer CNN, is now explained in more detail in the main text.

---

### Author Response · Authors · 2025-11-26
**Overview of main changes in revision**

We thank all referees for taking the time to review our work and for
their valuable comments. We'd like to draw your attention to several
important changes and additions in our revised version. Specifically:

1.  **Presentation**: The revised version includes a logical-flow
    diagram and a more detailed outline paragraph explaining how the
    different bounds and estimates interact. In addition, instead of
    describing three concrete examples, we now focus on the 3-layer FCN
    example and discuss that in depth. The remaining examples are
    presented in the appendix. Finally, we also emphasize that our
    layer-wise variational energy can be thought of as the excess weight
    decay due to feature learning, thus providing a more intuitive grasp
    of these quantities when the feature learning pattern involves
    well-defined circuits. Finally, we also made further clarifications
    of definitions and notations where necessary, and added further
    details to the derivation in the appendix.

2.  **Experiments**: The revised version includes two more experiments,
    which validate our heuristic and extend it to classification tasks,
    more complex activation functions, and architectures. Specifically,
    a *softmax attention block* predicted and demonstrated to have a
    sample complexity given by the square root of the context length and
    a *sparse-parity classification problem* involving ReLU activations,
    where we correctly predict the magnitude of specializing neurons.
    *In total, we now have 5 experimentally verified results predicted
    by our heuristics, 4 of which are new.* In particular, for
    2-Layer-Erf-FCN in standard scaling we predict number of
    specializing neurons and distribution of pre-activations, for
    2-Layer-Erf-CNN we verify GFL and $P=d^{3/4}$ sample complexity, for
    3-layer-Erf-FCNs we verify and predict a $N_1^{1/3}$ scaling of
    specializing neurons and $P=d$ sample complexity, for softmax
    attention block $P=\sqrt{L d^3}$ sample complexity, finally for
    $k=2$ sparse parity with cross-entropy loss for 2-layer-ReLU-FCN, we
    predict $O(1)$ specializing neurons with a $\sqrt{N}$ magnitude.

3.  **Theory**: The revised manuscript contains an extension of our
    sample complexity bound, and therefore our heuristic, to
    classification using cross-entropy loss.

---

### Meta-Review · Area_Chair_ZDwp · 2025-12-26

**Summary:**

This paper develops a variational, Bayesian-inspired framework to estimate how the sample complexity of feature learning scales with task and architecture. A key strength is that the authors directly target scaling laws for sample complexity, rather than solving specific cases: they obtain explicit n vs. N scaling predictions in several settings and show how these compose across layers. Using a bound on the variational “energy” and a set of scaling heuristics, they recover known results for 2-layer FCNs and CNNs with non-overlapping patches and extend them to harder cases such as 3-layer networks. The revised version also adds analyses and experiments for a softmax attention block and a ReLU sparse-parity classification setup, and clarifies the overall logical flow and notation.

Reviewers see the central question and the technical development as interesting and nontrivial, and two of them rate the paper marginally above the acceptance threshold. The main weaknesses are that (i) the framework is inherently heuristic and tied to a Bayesian/rare-event picture that is not a realistic posterior for highly overparameterized networks, (ii) empirical validation, while broader in the revision, is still limited to a small set of teacher–student-type tasks, and (iii) parts of the exposition remain dense and difficult to parse, especially for readers not already steeped in this line of work. In my view, the Bayesian perspective should be understood and presented explicitly as a modeling device to derive scaling predictions, not as a literal description of SGD-trained posteriors; within that scope, the direct scaling-focused predictions and the unifying treatment of several feature-learning patterns make this a valuable theoretical contribution that connects nicely to existing work. Overall, I view this as a solid, specialized theory paper that merits being on the record, and I lean toward acceptance, conditional on the camera-ready making the Bayesian status and heuristic nature of the framework more explicit and further polishing the exposition.

**Reviewer Concerns:**

Addressed:
The revision improves the presentation (clearer notation, schematic overview, better explanation of the logical flow), adds new experiments (attention block, ReLU sparse-parity classification), and clarifies how the variational energy, feature-propagation heuristics, and specialization patterns are derived and interpreted. Several specific technical ambiguities raised by reviewers (inequality direction, definitions, constants) were also clarified.

Still outstanding:
The framework remains largely heuristic and relies on a Bayesian/rare-event viewpoint whose relation to actual overparameterized training is not fully convincing. Empirical validation, although broader, is still limited in scope and mostly aligned with carefully chosen teacher–student tasks. Finally, parts of the theory (especially around three-layer models and pattern optimality) remain heavy and difficult to verify, and some key assumptions are more justified by analogy than by rigorous control.

**Reviewer Scores:**

eTgY (6) – New experiments + broader discussion likely reassuring -> probably unchanged (maybe +1).
tHzt (4) – Clarifications help, but core opacity remains -> likely unchanged.
9jc9 (4) – Added plots/explanations help somewhat, but likely still skeptical -> likely unchanged.
d7wZ (6) – Revisions strengthen case but heuristics remain -> likely unchanged (at most +1).

---

### Decision · Program_Chairs · 2026-01-26

Accept (Poster)